# AdaBelief Optimizer: Adapting Stepsizes by the Belief in Observed Gradients

**Juntang Zhuang**[1]; **Tommy Tang**[2]; **Yifan Ding**[3]; **Sekhar Tatikonda**[1]; **Nicha Dvornek**[1];
**Xenophon Papademetris**[1]; **James S. Duncan**[1]
[1] Yale University; [2] University of Illinois at Urbana-Champaign; [3] University of Central Florida
{j.zhuang;sekhar.tatikonda;nicha.dvornek;xenophon.papademetris;
james.duncan}@yale.edu; tommymt2@illinois.edu; yf.ding@knights.ucf.edu

## Abstract

Most popular optimizers for deep learning can be broadly categorized as adaptive methods (e.g. Adam) and accelerated schemes (e.g. stochastic gradient descent (SGD) with momentum). For many models such as convolutional neural networks (CNNs), adaptive methods typically converge faster but generalize worse compared to SGD; for complex settings such as generative adversarial networks (GANs), adaptive methods are typically the default because of their stability. We propose AdaBelief to simultaneously achieve three goals: fast convergence as in adaptive methods, good generalization as in SGD, and training stability. The intuition for AdaBelief is to adapt the stepsize according to the "belief" in the current gradient direction. Viewing the exponential moving average (EMA) of the noisy gradient as the prediction of the gradient at the next time step, if the observed gradient greatly deviates from the prediction, we distrust the current observation and take a small step; if the observed gradient is close to the prediction, we trust it and take a large step. We validate AdaBelief in extensive experiments, showing that it outperforms other methods with fast convergence and high accuracy on image classification and language modeling. Specifically, on ImageNet, AdaBelief achieves comparable accuracy to SGD. Furthermore, in the training of a GAN on Cifar10, AdaBelief demonstrates high stability and improves the quality of generated samples compared to a well-tuned Adam optimizer. Code is available at https://github.com/juntang-zhuang/Adabelief-Optimizer

## 1 Introduction

Modern neural networks are typically trained with first-order gradient methods, which can be broadly categorized into two branches: the accelerated stochastic gradient descent (SGD) family [1], such as Nesterov accelerated gradient (NAG) [2], SGD with momentum [3] and heavy-ball method (HB) [4]; and the adaptive learning rate methods, such as Adagrad [5], AdaDelta [6], RMSProp [7] and Adam [8]. SGD methods use a global learning rate for all parameters, while adaptive methods compute an individual learning rate for each parameter.

Compared to the SGD family, adaptive methods typically converge fast in the early training phases, but have poor generalization performance [9, 10]. Recent progress tries to combine the benefits of both, such as switching from Adam to SGD either with a hard schedule as in SWATS [11], or with a smooth transition as in AdaBound [12]. Other modifications of Adam are also proposed: AMSGrad [13] fixes the error in convergence analysis of Adam, Yogi [14] considers the effect of minibatch size, MSVAG [15] dissects Adam as sign update and magnitude scaling, RAdam [16] rectifies the variance of learning rate, Fromage [17] controls the distance in the function space, and AdamW [18] decouples weight decay from gradient descent. Although these modifications achieve better accuracy compared to Adam, their generalization performance is typically worse than SGD on large-scale

datasets such as ImageNet [19]; furthermore, compared with Adam, many optimizers are empirically unstable when training generative adversarial networks (GAN) [20].

To solve the problems above, we propose "AdaBelief", which can be easily modified from Adam. Denote the observed gradient at step $t$ as $g_t$ and its exponential moving average (EMA) as $m_t$. Denote the EMA of $g_t^2$ and $(g_t - m_t)^2$ as $v_t$ and $s_t$, respectively. $m_t$ is divided by $\sqrt{v_t}$ in Adam, while it is divided by $\sqrt{s_t}$ in AdaBelief. Intuitively, $\frac{1}{\sqrt{s_t}}$ is the "belief" in the observation: viewing $m_t$ as the prediction of the gradient, if $g_t$ deviates much from $m_t$, we have weak belief in $g_t$, and take a small step; if $g_t$ is close to the prediction $m_t$, we have a strong belief in $g_t$, and take a large step. We validate the performance of AdaBelief with extensive experiments. Our contributions can be summarized as:

- We propose AdaBelief, which can be easily modified from Adam without extra parameters. AdaBelief has three properties: (1) fast convergence as in adaptive gradient methods, (2) good generalization as in the SGD family, and (3) training stability in complex settings such as GAN.
- We theoretically analyze the convergence property of AdaBelief in both convex optimization and non-convex stochastic optimization.
- We validate the performance of AdaBelief with extensive experiments: AdaBelief achieves fast convergence as Adam and good generalization as SGD in image classification tasks on CIFAR and ImageNet; AdaBelief outperforms other methods in language modeling; in the training of a W-GAN [21], compared to a well-tuned Adam optimizer, AdaBelief significantly improves the quality of generated images, while several recent adaptive optimizers fail the training.

## 2 Methods

### 2.1 Details of AdaBelief Optimizer

**Notations** By the convention in [8], we use the following notations:

- $f(\theta) \in \mathbb{R}, \theta \in \mathbb{R}^d$: $f$ is the loss function to minimize, $\theta$ is the parameter in $\mathbb{R}^d$
- $\prod_{\mathcal{F},M}(y) = \operatorname{argmin}_{x \in \mathcal{F}} ||M^{1/2}(x - y)||$: projection of $y$ onto a convex feasible set $\mathcal{F}$
- $g_t$: the gradient and step $t$
- $m_t$: exponential moving average (EMA) of $g_t$
- $v_t, s_t$: $v_t$ is the EMA of $g_t^2$, $s_t$ is the EMA of $(g_t - m_t)^2$
- $\alpha, \epsilon$: $\alpha$ is the learning rate, default is $10^{-3}$; $\epsilon$ is a small number, typically set as $10^{-8}$
- $\beta_1, \beta_2$: smoothing parameters, typical values are $\beta_1 = 0.9, \beta_2 = 0.999$
- $\beta_{1t}, \beta_{2t}$ are the momentum for $m_t$ and $v_t$ respectively at step $t$, and typically set as constant (e.g. $\beta_{1t} = \beta_1, \beta_{2t} = \beta_2, \forall t \in \{1, 2, ...T\}$

| **Algorithm 1:** Adam Optimizer | **Algorithm 2:** AdaBelief Optimizer |
|---|---|
| **Initialize** $\theta_0, m_0 \leftarrow 0$ , $v_0 \leftarrow 0, t \leftarrow 0$ | **Initialize** $\theta_0, m_0 \leftarrow 0$ , $s_0 \leftarrow 0, t \leftarrow 0$ |
| **While** $\theta_t$ not converged | **While** $\theta_t$ not converged |
| $\quad t \leftarrow t + 1$ | $\quad t \leftarrow t + 1$ |
| $\quad g_t \leftarrow \nabla_\theta f_t(\theta_{t-1})$ | $\quad g_t \leftarrow \nabla_\theta f_t(\theta_{t-1})$ |
| $\quad m_t \leftarrow \beta_1 m_{t-1} + (1 - \beta_1)g_t$ | $\quad m_t \leftarrow \beta_1 m_{t-1} + (1 - \beta_1)g_t$ |
| $\quad v_t \leftarrow \beta_2 v_{t-1} + (1 - \beta_2)g_t^2$ | $\quad s_t \leftarrow \beta_2 s_{t-1} + (1-\beta_2)(g_t - m_t)^2 + \epsilon$ |
| $\quad$**Bias Correction** | $\quad$**Bias Correction** |
| $\quad\quad \widehat{m_t} \leftarrow \frac{m_t}{1-\beta_1^t}, \widehat{v_t} \leftarrow \frac{v_t}{1-\beta_2^t}$ | $\quad\quad \widehat{m_t} \leftarrow \frac{m_t}{1-\beta_1^t}, \widehat{s_t} \leftarrow \frac{s_t}{1-\beta_2^t}$ |
| $\quad$**Update** | $\quad$**Update** |
| $\quad\quad \theta_t \leftarrow \prod_{\mathcal{F},\sqrt{\widehat{v_t}}}\left(\theta_{t-1} - \frac{\alpha \widehat{m_t}}{\sqrt{\widehat{v_t}}+\epsilon}\right)$ | $\quad\quad \theta_t \leftarrow \prod_{\mathcal{F},\sqrt{\widehat{s_t}}}\left(\theta_{t-1} - \frac{\alpha \widehat{m_t}}{\sqrt{\widehat{s_t}}+\epsilon}\right)$ |

**Comparison with Adam** Adam and AdaBelief are summarized in Algo. 1 and Algo. 2, where all operations are element-wise, with differences marked in blue. Note that no extra parameters are introduced in AdaBelief. Specifically, in Adam, the update direction is $m_t/\sqrt{v_t}$, where $v_t$ is the EMA of $g_t^2$; in AdaBelief, the update direction is $m_t/\sqrt{s_t}$, where $s_t$ is the EMA of $(g_t - m_t)^2$. Intuitively, viewing $m_t$ as the prediction of $g_t$, AdaBelief takes a large step when observation $g_t$ is close to prediction $m_t$, and a small step when the observation greatly deviates from the prediction. $\widehat{\cdot}$ represents bias-corrected value. Note that an extra $\epsilon$ is added to $s_t$ during bias-correction, in order to

Table 1: Comparison of optimizers in various cases in Fig. 1. "S" and "L" represent "small" and "large" stepsize, respectively. $|\Delta\theta_t|_{ideal}$ is the stepsize of an ideal optimizer. Note that only AdaBelief matches the behaviour of an ideal optimizer in all three cases.

| | Case 1 | | | Case 2 | | | Case 3 | | |
|---|---|---|---|---|---|---|---|---|---|
| $|g_t|, v_t$ | S | | | L | | | L | | |
| $|g_t - g_{t-1}|, s_t$ | S | | | L | | | S | | |
| $|\Delta\theta_t|_{ideal}$ | L | | | S | | | L | | |
| $|\Delta\theta_t|$ | SGD | Adam | AdaBelief | SGD | Adam | AdaBelief | SGD | Adam | AdaBelief |
| | S | L | L | L | S | S | L | S | L |

better match the assumption that $s_t$ is bouded below (the lower bound is at leat $\epsilon$). For simplicity, we omit the bias correction step in theoretical analysis.

## 2.2 Intuitive explanation for benefits of AdaBelief

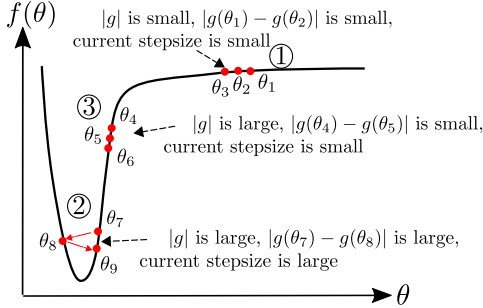

Figure 1: An ideal optimizer considers curvature of the loss function, instead of taking a large (small) step where the gradient is large (small) [22].

**AdaBelief uses curvature information** Update formulas for SGD, Adam and AdaBelief are:

$$\Delta\theta_t^{SGD} = -\alpha m_t, \quad \Delta\theta_t^{Adam} = -\alpha m_t / \sqrt{v_t},$$
$$\Delta\theta_t^{AdaBelief} = -\alpha m_t / \sqrt{s_t} \qquad (1)$$

Note that we name $\alpha$ as the "learning rate" and $|\Delta\theta_t^i|$ as the "stepsize" for the $i$th parameter. With a 1D example in Fig. 1, we demonstrate that AdaBelief uses the curvature of loss functions to improve training as summarized in Table 1, with a detailed description below:

(1) In region ① in Fig. 1, the loss function is flat, hence the gradient is close to 0. In this case, an ideal optimizer should take a large stepsize. The stepsize of SGD is proportional to the EMA of the gradient, hence is small in this case; while both Adam and AdaBelief take a large stepsize, because the denominator ($\sqrt{v_t}$ and $\sqrt{s_t}$) is a small value.

(2) In region ②, the algorithm oscillates in a "steep and narrow" valley, hence both $|g_t|$ and $|g_t - g_{t-1}|$ is large. An ideal optimizer should decrease its stepsize, while SGD takes a large step (proportional to $m_t$). Adam and AdaBelief take a small step because the denominator ($\sqrt{s_t}$ and $\sqrt{v_t}$) is large.

(3) In region ③, we demonstrate AdaBelief's advantage over Adam in the "large gradient, small curvature" case. In this case, $|g_t|$ and $v_t$ are large, but $|g_t - g_{t-1}|$ and $s_t$ are small; this could happen because of a small learning rate $\alpha$. In this case, an ideal optimizer should increase its stepsize. SGD uses a large stepsize ($\sim \alpha|g_t|$); in Adam, the denominator $\sqrt{v_t}$ is large, hence the stepsize is small; in AdaBelief, denominator $\sqrt{s_t}$ is small, hence the stepsize is large as in an ideal optimizer.

To sum up, AdaBelief scales the update direction by the change in gradient, which is related to the Hessian. Therefore, AdaBelief considers curvature information and performs better than Adam.

**AdaBelief considers the sign of gradient in denominator** We show the advantages of AdaBelief with a 2D example in this section, which gives us more intuition for high dimensional cases. In Fig. 2, we consider the loss function: $f(x, y) = |x| + |y|$. Note that in this simple problem, the gradient in each axis can only take $\{1, -1\}$. Suppose the start point is near the $x-$axis, e.g. $y_0 \approx 0, x_0 \ll 0$. Optimizers will oscillate in the $y$ direction, and keep increasing in the $x$ direction. Suppose the algorithm runs for a long time ($t$ is large), so the bias of EMA ($\beta_1^t \mathbb{E}g_t$) is small:

$$m_t = EMA(g_0, g_1, ...g_t) \approx \mathbb{E}(g_t), \quad m_{t,x} \approx \mathbb{E}g_{t,x} = 1, \quad m_{t,y} \approx \mathbb{E}g_{t,y} = 0 \qquad (2)$$
$$v_t = EMA(g_0^2, g_1^2, ...g_t^2) \approx \mathbb{E}(g_t^2), \quad v_{t,x} \approx \mathbb{E}g_{t,x}^2 = 1, \quad v_{t,y} \approx \mathbb{E}g_{t,y}^2 = 1. \qquad (3)$$



| | Step | 1 | 2 | 3 | 4 | 5 |
|---|---|---|---|---|---|---|
| | $g_x$ | 1 | 1 | 1 | 1 | 1 |
| | $g_y$ | -1 | 1 | -1 | 1 | -1 |
| Adam | $v_x$ | 1 | 1 | 1 | 1 | 1 |
| | $v_y$ | 1 | 1 | 1 | 1 | 1 |
| AdaBelief | $s_x$ | 0 | 0 | 0 | 0 | 0 |
| | $s_y$ | 1 | 1 | 1 | 1 | 1 |

Figure 2: *Left:* Consider $f(x, y) = |x| + |y|$. Blue vectors represent the gradient, and the cross represents the optimal point. The optimizer oscillates in the $y$ direction, and keeps moving forward in the $x$ direction. *Right:* Optimization process for the example on the left. Note that denominator $\sqrt{v_{t,x}} = \sqrt{v_{t,y}}$ for Adam, hence the same stepsize in $x$ and $y$ direction; while $\sqrt{s_{t,x}} < \sqrt{s_{t,y}}$, hence AdaBelief takes a large step in the $x$ direction, and a small step in the $y$ direction.

In practice, the bias correction step will further reduce the error between the EMA and its expectation if $g_t$ is a stationary process [8]. Note that:

$$s_t = EMA\big((g_0 - m_0)^2, ...(g_t - m_t)^2\big) \approx \mathbb{E}\big[(g_t - \mathbb{E}g_t)^2\big] = \mathbf{Var}g_t, \ \ s_{t,x} \approx 0, \ \ s_{t,y} \approx 1 \quad (4)$$

An example of the analysis above is summarized in Fig. 2. From Eq. 3 and Eq. 4, note that in Adam, $v_x = v_y$; this is because the update of $v_t$ only uses the amplitude of $g_t$ and ignores its sign, hence the stepsize for the $x$ and $y$ direction is the same $1/\sqrt{v_{t,x}} = 1/\sqrt{v_{t,y}}$. AdaBelief considers both the magnitude and sign of $g_t$, and $1/\sqrt{s_{t,x}} \gg 1/\sqrt{s_{t,y}}$, hence takes a large step in the $x$ direction and a small step in the $y$ direction, which matches the behaviour of an ideal optimizer.

**Update direction in Adam is close to "sign descent" in low-variance case**   In this section, we demonstrate that when the gradient has low variance, the update direction in Adam is close to "sign descent", hence deviates from the gradient. This is also mentioned in [15].

Under the following assumptions: (1) assume $g_t$ is drawn from a stationary distribution, hence after bias correction, $\mathbb{E}v_t = (\mathbb{E}g_t)^2 + \mathbf{Var}g_t$. (2) low-noise assumption, assume $(\mathbb{E}g_t)^2 \gg \mathbf{Var}g_t$, hence we have $\mathbb{E}g_t/\sqrt{\mathbb{E}v_t} \approx \mathbb{E}g_t/\sqrt{(\mathbb{E}g_t)^2} = sign(\mathbb{E}g_t)$. (3) low-bias assumption, assume $\beta_1^t$ ($\beta_1$ to the power of $t$) is small, hence $m_t$ as an estimator of $\mathbb{E}g_t$ has a small bias $\beta_1^t\mathbb{E}g_t$. Then

$$\Delta\theta_t^{Adam} = -\alpha\frac{m_t}{\sqrt{v_t}+\epsilon} \approx -\alpha\frac{\mathbb{E}g_t}{\sqrt{(\mathbb{E}g_t)^2+\mathbf{Var}g_t}+\epsilon} \approx -\alpha\frac{\mathbb{E}g_t}{||\mathbb{E}g_t||} = -\alpha\,\text{sign}(\mathbb{E}g_t) \quad (5)$$

In this case, Adam behaves like a "sign descent"; in 2D cases the update is $\pm 45°$ to the axis, hence deviates from the true gradient direction. The "sign update" effect might cause the generalization gap between adaptive methods and SGD (e.g. on ImageNet) [23, 9]. For AdaBelief, when the variance of $g_t$ is the same for all coordinates, the update direction matches the gradient direction; when the variance is not uniform, AdaBelief takes a small (large) step when the variance is large (small).

**Numerical experiments**   In this section, we validate intuitions in Sec. 2.2. Examples are shown in Fig. 3, and we refer readers to more *video examples*[1] for better visualization. In all examples, compared with SGD with momentum and Adam, AdaBelief reaches the optimal point at the fastest speed. Learning rate is $\alpha = 10^{-3}$ for all optimizers. For all examples except Fig. 3(d), we set the parameters of AdaBelief to be the same as the default in Adam [8], $\beta_1 = 0.9, \beta_2 = 0.999, \epsilon = 10^{-8}$, and set momentum as 0.9 for SGD. For Fig. 3(d), to match the assumption in Sec. 2.2, we set $\beta_1 = \beta_2 = 0.3$ for both Adam and AdaBelief, and set momentum as 0.3 for SGD.

(a) Consider the loss function $f(x, y) = |x| + |y|$ and a starting point near the $x$ axis. This setting corresponds to Fig. 2. Under the same setting, AdaBelief takes a large step in the $x$ direction, and a small step in the $y$ direction, validating our analysis. More examples such as $f(x, y) = |x|/10 + |y|$ are in the supplementary videos.

(b) For an inseparable $L_1$ loss, AdaBelief outperforms other methods under the same setting.

(c) For an inseparable $L_2$ loss, AdaBelief outperforms other methods under the same setting.

(d) We set $\beta_1 = \beta_2 = 0.3$ for Adam and AdaBelief, and set momentum as 0.3 in SGD. This corresponds to settings of Eq. 5. For the loss $f(x, y) = |x|/10 + |y|$, $g_t$ is a constant for a large region, hence $||\mathbb{E}g_t|| \gg \mathbf{Var}g_t$. As mentioned in [8], $\mathbb{E}m_t = (1 - \beta^t)\mathbb{E}g_t$, hence a

[1] https://www.youtube.com/playlist?list=PL7KkG3n9bER6YmMLrKJ5wocjlvP7aWoOu

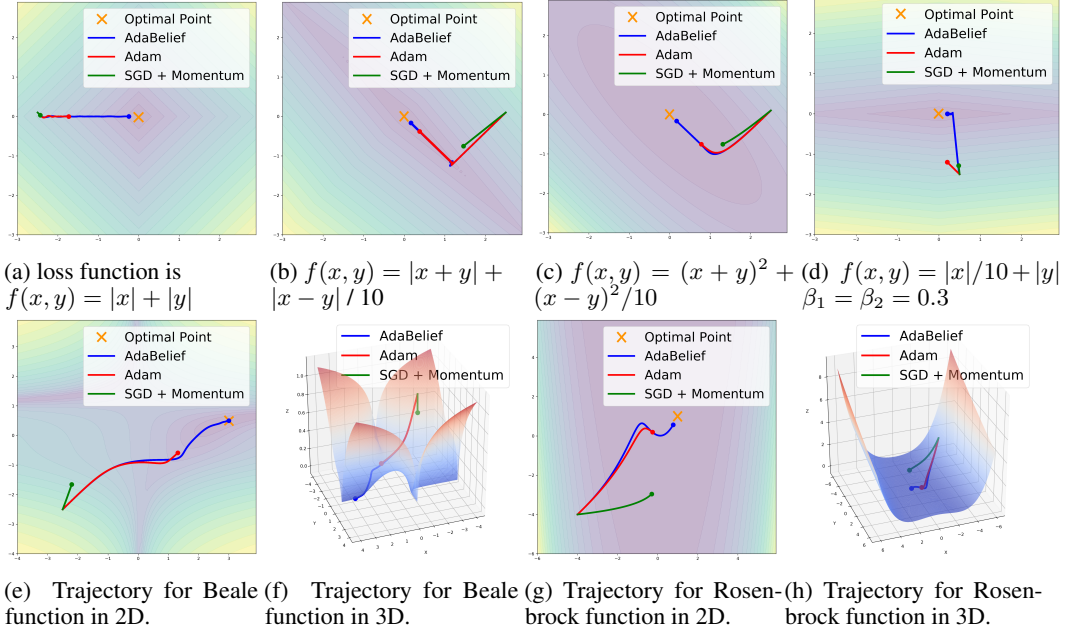

(a) loss function is $f(x,y) = |x| + |y|$

(b) $f(x,y) = |x+y| + |x-y| / 10$

(c) $f(x,y) = (x+y)^2 + (x-y)^2/10$

(d) $f(x,y) = |x|/10 + |y|$ $\beta_1 = \beta_2 = 0.3$

(e) Trajectory for Beale function in 2D.

(f) Trajectory for Beale function in 3D.

(g) Trajectory for Rosenbrock function in 2D.

(h) Trajectory for Rosenbrock function in 3D.

Figure 3: Trajectories of SGD, Adam and AdaBelief. AdaBelief reaches optimal point (marked as orange cross in 2D plots) the fastest in all cases. We refer readers to *video examples*.

smaller $\beta$ decreases $||m_t - \mathbb{E}g_t||$ faster to 0. Adam behaves like a sign descent ($45°$ to the axis), while AdaBelief and SGD update in the direction of the gradient.

(e)-(f) Optimization trajectory under default setting for the Beale [24] function in 2D and 3D.

(g)-(h) Optimization trajectory under default setting for the Rosenbrock [25] function.

**Above cases occur frequently in deep learning** Although the above cases are simple, they give hints to local behavior of optimizers in deep learning, and we expect them to occur frequently in deep learning. Hence, we expect AdaBelief to outperform Adam in *general cases*. Other works in the literature [13, 12] claim advantages over Adam, but are typically substantiated with *carefully-constructed examples*. Note that most deep networks use ReLU activation [26], which behaves like an absolute value function as in Fig. 3(a). Considering the interaction between neurons, most networks behave like case Fig. 3(b), and typically are ill-conditioned (the weight of some parameters are far larger than others) as in the figure. Considering a smooth loss function such as cross entropy or a smooth activation, this case is similar to Fig. 3(c). The case with Fig. 3(d) requires $|m_t| \approx |\mathbb{E}g_t| \gg \mathbf{Var}g_t$, and this typically occurs at the late stages of training, where the learning rate $\alpha$ is decayed to a small value, and the network reaches a stable region.

## 2.3 Convergence analysis in convex and non-convex optimization

Similar to [13, 12, 27], for simplicity, we omit the de-biasing step (analysis applicable to de-biased version). Proof for convergence in convex and non-convex cases is in the appendix.

**Optimization problem** For deterministic problems, the problem to be optimized is $\min_{\theta \in \mathcal{F}} f(\theta)$; for online optimization, the problem is $\min_{\theta \in \mathcal{F}} \sum_{t=1}^{T} f_t(\theta)$, where $f_t$ can be interpreted as loss of the model with the chosen parameters in the $t$-th step.

**Theorem 2.1.** *(Convergence in convex optimization) Let $\{\theta_t\}$ and $\{s_t\}$ be the sequence obtained by AdaBelief, let $0 \leq \beta_2 < 1, \alpha_t = \frac{\alpha}{\sqrt{t}}, \beta_{11} = \beta_1, 0 \leq \beta_{1t} \leq \beta_1 < 1, s_t \leq s_{t+1}, \forall t \in [T]$. Let $\theta \in \mathcal{F}$, where $\mathcal{F} \subset \mathbb{R}^d$ is a convex feasible set with bounded diameter $D_\infty$. Assume $f(\theta)$ is a convex function and $||g_t||_\infty \leq G_\infty/2$ (hence $||g_t - m_t||_\infty \leq G_\infty$) and $s_{t,i} \geq c > 0, \forall t \in [T], \theta \in \mathcal{F}$. Denote the*

*optimal point as $\theta^*$. For $\theta_t$ generated with AdaBelief, we have the following bound on the regret:*

$$\sum_{t=1}^{T}[f_t(\theta_t) - f_t(\theta^*)] \leq \frac{D_\infty^2 \sqrt{T}}{2\alpha(1-\beta_1)}\sum_{i=1}^{d} s_{T,i}^{1/2} + \frac{(1+\beta_1)\alpha\sqrt{1+\log T}}{2\sqrt{c}(1-\beta_1)^3}\sum_{i=1}^{d}\left\|g_{1:T,i}^2\right\|_2 + \frac{D_\infty^2}{2(1-\beta_1)}\sum_{t=1}^{T}\sum_{i=1}^{d}\frac{\beta_{1t}s_{t,i}^{1/2}}{\alpha_t}$$

**Corollary 2.1.1.** *Suppose $\beta_{1,t} = \beta_1\lambda^t$, $0 < \lambda < 1$ in Theorem (2.1), then we have:*

$$\sum_{t=1}^{T}[f_t(\theta_t) - f_t(\theta^*)] \leq \frac{D_\infty^2 \sqrt{T}}{2\alpha(1-\beta_1)}\sum_{i=1}^{d} s_{T,i}^{1/2} + \frac{(1+\beta_1)\alpha\sqrt{1+\log T}}{2\sqrt{c}(1-\beta_1)^3}\sum_{i=1}^{d}\left\|g_{1:T,i}^2\right\|_2 + \frac{D_\infty^2\beta_1 G_\infty}{2(1-\beta_1)(1-\lambda)^2\alpha}$$

For the convex case, Theorem 2.1 implies the regret of AdaBelief is upper bounded by $O(\sqrt{T})$. Conditions for Corollary 2.1.1 can be relaxed to $\beta_{1,t} = \beta_1/t$ as in [13], which still generates $O(\sqrt{T})$ regret. Similar to Theorem 4.1 in [8] and corollary 1 in [13], where the term $\sum_{i=1}^{d} v_{T,i}^{1/2}$ exists, we have $\sum_{i=1}^{d} s_{T,i}^{1/2}$. Without further assumption, $\sum_{i=1}^{d} s_{T,i}^{1/2} < dG_\infty$ since $\|g_t - m_t\|_\infty < G_\infty$ as assumed in Theorem 2.1, and $dG_\infty$ is constant. The literature [8, 13, 5] exerts a stronger assumption that $\sum_{i=1}^{d} \sqrt{T}v_{T,i}^{1/2} \ll dG_\infty\sqrt{T}$. Our assumption could be similar or weaker, because $\mathbb{E}s_t = \mathrm{Var}g_t \leq \mathbb{E}g_t^2 = \mathbb{E}v_t$, then we get better regret than $O(\sqrt{T})$.

**Theorem 2.2.** *(Convergence for non-convex stochastic optimization) Under the assumptions:*

- *$f$ is differentiable; $\|\nabla f(x) - \nabla f(y)\| \leq L\|x - y\|$, $\forall x, y$; $f$ is also lower bounded.*
- *The noisy gradient is unbiased, and has independent noise, i.e. $g_t = \nabla f(\theta_t) + \zeta_t, \mathbb{E}\zeta_t = 0, \zeta_t \perp \zeta_j, \forall t, j \in \mathbb{N}, t \neq j$.*
- *At step $t$, the algorithm can access a bounded noisy gradient, and the true gradient is also bounded. i.e. $\|\nabla f(\theta_t)\| \leq H, \|g_t\| \leq H, \forall t > 1$.*

*Assume $\min_{j\in[d]}(s_1)_j \geq c > 0$, noise in gradient has bounded variance, $\mathrm{Var}(g_t) = \sigma_t^2 \leq \sigma^2, s_t \leq s_{t+1}, \forall t \in \mathbb{N}$, then the proposed algorithm satisfies:*

$$\min_{t\in[T]} \mathbb{E}\left\|\nabla f(\theta_t)\right\|^2 \leq \frac{H}{\sqrt{T}\alpha}\left[\frac{C_1\alpha^2(H^2+\sigma^2)(1+\log T)}{c} + C_2\frac{d\alpha}{\sqrt{c}} + C_3\frac{d\alpha^2}{c} + C_4\right]$$

*as in [27], $C_1, C_2, C_3$ are constants independent of $d$ and $T$, and $C_4$ is a constant independent of $T$.*

**Corollary 2.2.1.** *If $c > C_1H$ and assumptions for Theorem 2.2 are satisfied, we have:*

$$\frac{1}{T}\sum_{t=1}^{T}\mathbb{E}\left[\alpha_t^2\left\|\nabla f(\theta_t)\right\|^2\right] \leq \frac{1}{T}\frac{1}{\frac{1}{H}-\frac{C_1}{c}}\left[\frac{C_1\alpha^2\sigma^2}{c}\left(1+\log T\right) + C_2\frac{d\alpha}{\sqrt{c}} + C_3\frac{d\alpha^2}{c} + C_4\right]$$

Theorem 2.2 implies the convergence rate for AdaBelief in the non-convex case is $O(\log T/\sqrt{T})$, which is similar to Adam-type optimizers [13, 27]. Note that regret bounds are derived in the *worst possible case*, while empirically AdaBelief outperforms Adam mainly because the cases in Sec. 2.2 occur more frequently. It is possible that the above bounds are loose. Also note that we assume $s_t \leq s_{t+1}$, in code this requires to use element wise maximum between $s_t$ and $s_{t+1}$ in the denominator.

# 3 Experiments

We performed extensive comparisons with other optimizers, including SGD [3], AdaBound [12], Yogi [14], Adam [8], MSVAG [15], RAdam [16], Fromage [17] and AdamW [18]. The experiments include: (a) image classification on Cifar dataset [28] with VGG [29], ResNet [30] and DenseNet [31], and image recognition with ResNet on ImageNet [32]; (b) language modeling with LSTM [33] on Penn TreeBank dataset [34]; (c) wasserstein-GAN (WGAN) [21] on Cifar10 dataset. We emphasize (c) because prior work focuses on convergence and accuracy, yet neglects training stability.

**Hyperparameter tuning** We performed a careful hyperparameter tuning in experiments. On image classification and language modeling we use the following:

- *AdaBelief:* We use the default parameters of Adam: $\beta_1 = 0.9, \beta_2 = 0.999, \epsilon = 10^{-8}, \alpha = 10^{-3}$.
- *SGD, Fromage:* We set the momentum as $0.9$, which is the default for many networks such as ResNet [30] and DenseNet[31]. We search learning rate among $\{10.0, 1.0, 0.1, 0.01, 0.001\}$.

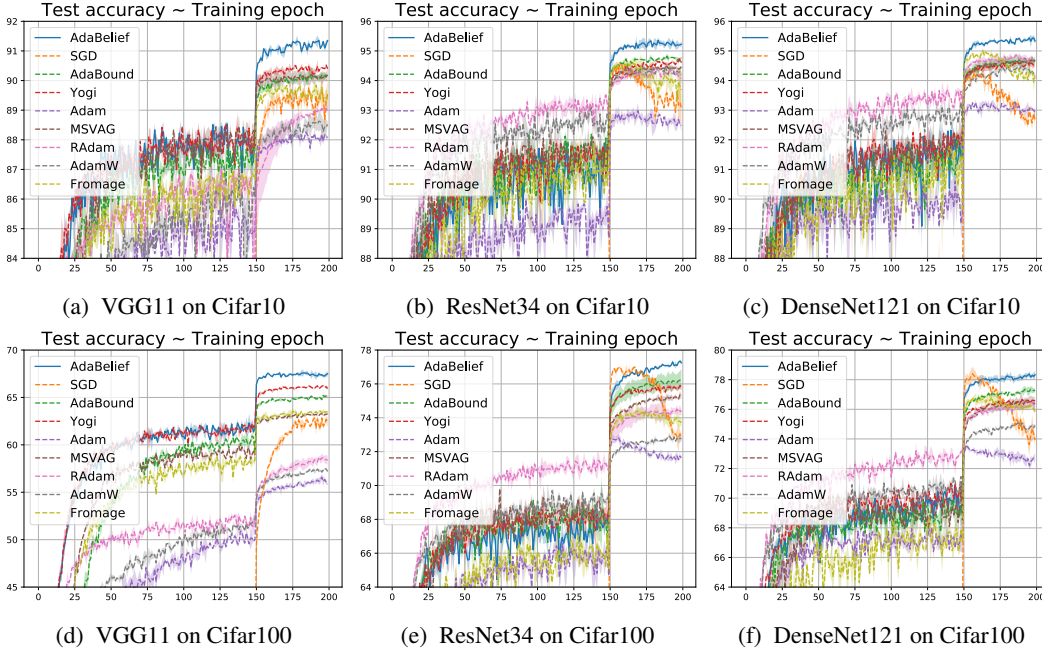

Figure 4: Test accuracy ($[\mu \pm \sigma]$) on Cifar. Code modified from official implementation of AdaBound.

Table 2: Top-1 accuracy of ResNet18 on ImageNet. † is reported in [35], ‡ is reported in [16]

| AdaBelief | SGD | AdaBound | Yogi | Adam | MSVAG | RAdam | AdamW |
|-----------|-----|----------|------|------|-------|-------|-------|
| **70.08** | 70.23† | 68.13† | 68.23† | 63.79† (66.54‡) | 65.99 | 67.62‡ | 67.93† |

● *Adam, Yogi, RAdam, MSVAG, AdaBound:* We search for optimal $\beta_1$ among $\{0.5, 0.6, 0.7, 0.8, 0.9\}$, search for $\alpha$ as in SGD, and set other parameters as their own default values in the literature.
● *AdamW:* We use the same parameter searching scheme as Adam. For other optimizers, we set the weight decay as $5 \times 10^{-4}$; for AdamW, since the optimal weight decay is typically larger [18], we search weight decay among $\{10^{-4}, 5 \times 10^{-4}, 10^{-3}, 10^{-2}\}$.
For the training of a GAN, we set $\beta_1 = 0.5, \epsilon = 10^{-12}$ for AdaBelief in a small GAN with vanilla CNN generator, and use $\epsilon = 10^{-16}$ for a larger spectral normalization GAN (SN-GAN) with a ResNet generator; for other methods, we search for $\beta_1$ among $\{0.5, 0.6, 0.7, 0.8, 0.9\}$, and search for $\epsilon$ among $\{10^{-3}, 10^{-5}, 10^{-8}, 10^{-10}, 10^{-12}\}$. We set learning rate as $2 \times 10^{-4}$ for all methods. Note that the recommended parameters for Adam [36] and for RMSProp [37] are within the search range.

**CNNs on image classification**   We experiment with VGG11, ResNet34 and DenseNet121 on Cifar10 and Cifar100 dataset. We use the *official implementation* of AdaBound, hence achieved an *exact replication* of [12]. For each optimizer, we search for the optimal hyperparameters, and report the mean and standard deviation of test-set accuracy (under optimal hyperparameters) for 3 runs with random initialization. As Fig. 4 shows, AdaBelief achieves fast convergence as in adaptive methods such as Adam while achieving better accuracy than SGD and other methods.

We then train a ResNet18 on ImageNet, and report the accuracy on the validation set in Table 2. Due to the heavy computational burden, we could not perform an extensive hyperparameter search; instead, we report the result of AdaBelief with the default parameters of Adam ($\beta_1 = 0.9, \beta_2 = 0.999, \epsilon = 10^{-8}$) and decoupled weight decay as in [16, 18]; for other optimizers, we report the *best result in the literature*. AdaBelief outperforms other adaptive methods and achieves comparable accuracy to SGD (70.08 v.s. 70.23), which closes the generalization gap between adaptive methods and SGD. Experiments validate the fast convergence and good generalization performance of AdaBelief.

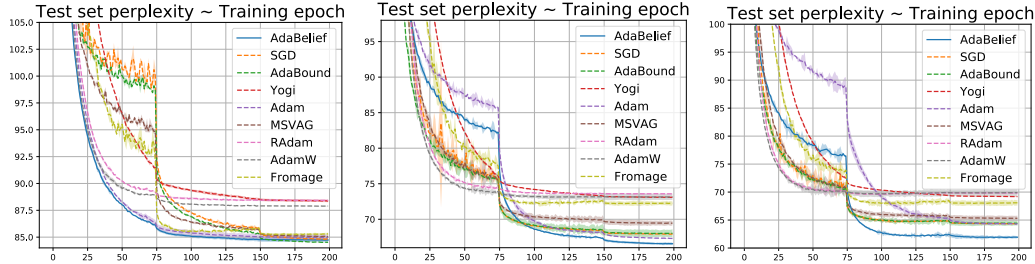

Figure 5: Left to right: perplexity ($[\mu \pm \sigma]$) on Penn Treebank for 1,2,3-layer LSTM. **Lower** is better.

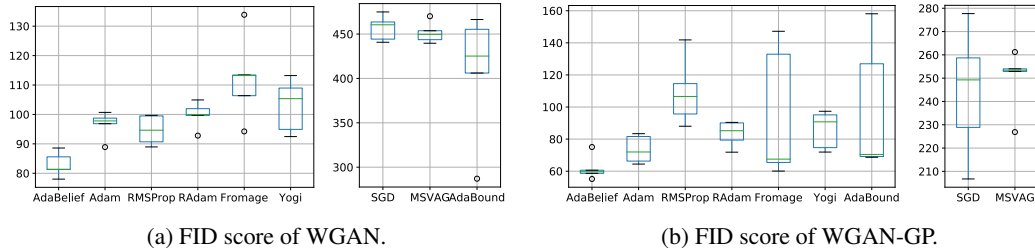

(a) FID score of WGAN.                                    (b) FID score of WGAN-GP.

Figure 6: FID score of WGAN and WGAN-GP using a vanilla CNN generator on Cifar10. **Lower** is better. For each model, successful and failed optimizers are shown in the left and right respectively, with different ranges in y value.

**LSTM on language modeling**   We experiment with LSTM on the Penn TreeBank dataset [34], and report the perplexity (lower is better) on the test set in Fig. 5. We report the mean and standard deviation across 3 runs. For both 2-layer and 3-layer LSTM models, AdaBelief achieves the lowest perplexity, validating its fast convergence as in adaptive methods and good accuracy. For the 1-layer model, the performance of AdaBelief is close to other optimizers.

**Generative adversarial networks**   Stability of optimizers is important in practice such as training of GANs, yet recently proposed optimizers often lack experimental validations. The training of a GAN alternates between generator and discriminator in a mini-max game, and is typically unstable [20]; SGD often generates mode collapse, and adaptive methods such as Adam and RMSProp are recommended in practice [38, 37, 39]. Therefore, training of GANs is a good test for the stability.

We experiment with one of the most widely used models, the Wasserstein-GAN (WGAN) [21] and the improved version with gradient penalty (WGAN-GP) [37] using a small model with vanilla CNN generator. Using each optimizer, we train the model for 100 epochs, generate 64,000 fake images from noise, and compute the Frechet Inception Distance (FID) [40] between the fake images and real dataset (60,000 real images). FID score captures both the quality and diversity of generated images and is widely used to assess generative models (lower FID is better). For each optimizer, under its optimal hyperparameter settings, we perform 5 runs of experiments, and report the results in Fig. 6 and Fig. 7. AdaBelief significantly outperforms other optimizers, and achieves the lowest FID score.

Besides the small model above, we also experiment with a large model using a ResNet generator and spectral normalization in the discriminator (SN-GAN). Results are summarized in Table. 3. Compared with a vanilla GAN, all FID scores are lower because the SN-GAN is more advanced. Compared with other optimizers, AdaBelief achieves the lowest FID with both large and small GANs.

**Remarks**   Recent research on optimizers tries to combine the fast convergence of adaptive methods with high accuracy of SGD. AdaBound [12] achieves this goal on Cifar, yet its performance on ImageNet is still inferior to SGD [35]. Padam [35] closes this generalization gap on ImageNet; writing the update as $\theta_{t+1} = \theta_t - \alpha m_t / v_t^p$, SGD sets $p = 0$, Adam sets $p = 0.5$, and Padam searches $p$ between 0 and 0.5 (outside this region Padam diverges [35, 41]). Intuitively, compared to Adam, by using a smaller $p$, Padam sacrifices the adaptivity for better generalization as in SGD; however, without good adaptivity, Padam loses training stability. As in Table 4, compared with Padam, AdaBelief achieves a much lower FID score in the training of GAN, meanwhile achieving

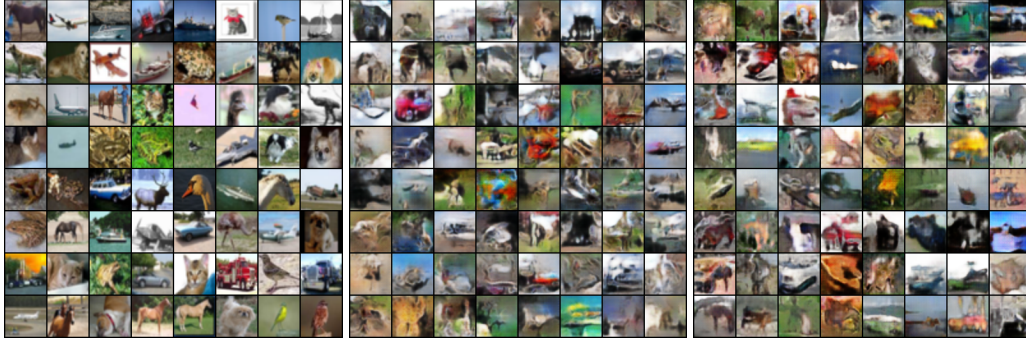

Figure 7: Left to right: real images, samples from WGAN, WGAN-GP (both trained by AdaBelief).

Table 3: FID (lower is better) of a SN-GAN with ResNet generator on Cifar10.

| AdaBelief | RAdam | RMSProp | Adam | Fromage | Yogi | SGD | MSVAG | AdaBound |
|-----------|-------|---------|------|---------|------|-----|-------|----------|
| **12.52 ± 0.16** | 12.70 ± 0.12 | 13.13 ± 0.12 | 13.05 ± 0.19 | 42.75 ± 0.15 | 14.25 ± 0.15 | 49.70 ± 0.41 | 48.35 ± 5.44 | 55.65 ± 2.15 |

Table 4: Comparison of AdaBelief and Padam. Higher Acc (lower FID) is better. ‡ is from [35].

| | AdaBelief | Padam | | | | | | |
|---|-----------|-------------|------|------|------|------|-------|-------------|
| | | p=1/2 (Adam) | p=2/5 | p=1/4 | p=1/5 | p=1/8 | p=1/16 | p = 0 (SGD) |
| ImageNet Acc | 70.08 | 63.79‡ | - | - | - | 70.07‡ | - | **70.23** ‡ |
| FID (WGAN) | **83.0± 4.1** | 96.6±4.5 | 97.5±2.8 | 426.4±49.6 | 401.5±33.2 | 328.1±37.2 | 362.6±43.9 | 469.3 ± 7.9 |
| FID (WGAN-GP) | **61.8± 7.7** | 73.5±8.7 | 87.1±6.0 | 155.1±23.8 | 167.3±27.6 | 203.6±18.9 | 228.5±25.8 | 244.3± 27.4 |

slightly higher accuracy on ImageNet classification. Furthermore, AdaBelief has the same number of parameters as Adam, while Padam has one more parameter hence is harder to tune.

**Extra experiments**     We conducted extra experiments, including Transformer models, reinforcement learning and object detection, and AdaBelief achieved better results than other optimizers. For details and code to reproduce results, please refer to our github page.

# 4     Related works

This work considers the update step in first-order methods. Other directions include Lookahead [42] which updates "fast" and "slow" weights separately, and is a wrapper that can combine with other optimizers; variance reduction methods [43, 44, 45] which reduce the variance in gradient; and LARS [46] which uses a layer-wise learning rate scaling. AdaBelief can be combined with these methods. Other variants of Adam have also been proposed (e.g. NosAdam [47], Sadam [48] and Adax [49]).

Besides first-order methods, second-order methods (e.g. Newton's method [50], Quasi-Newton method and Gauss-Newton method [51, 52, 51], L-BFGS [53], Natural-Gradient [54, 55], Conjugate-Gradient [56]) are widely used in conventional optimization. Hessian-free optimization (HFO) [57] uses second-order methods to train neural networks. Second-order methods typically use curvature information and are invariant to scaling [58] but have heavy computational burden, and hence are not widely used in deep learning.

# 5     Conclusion

We propose the AdaBelief optimizer, which adaptively scales the stepsize by the difference between predicted gradient and observed gradient. To our knowledge, AdaBelief is the first optimizer to achieve three goals simultaneously: fast convergence as in adaptive methods, good generalization as in SGD, and training stability in complex settings such as GANs. Furthermore, Adabelief has the same parameters as Adam, hence is easy to tune. We validate the benefits of AdaBelief with intuitive examples, theoretical convergence analysis in both convex and non-convex cases, and extensive experiments on real-world datasets.

## Broader Impact

Optimization is at the core of modern machine learning, and numerous efforts have been put into it. To our knowledge, AdaBelief is the first optimizer to achieve fast speed, good generalization and training stability. Adabelief can be used for the training of all models that can numerically estimate parameter gradients, hence can boost the development and application of deep learning models. This work mainly focuses on the theory part, and the social impact is mainly determined by each application rather than by optimizer.

## Acknowledgments and Disclosure of Funding

This research is supported by NIH grant R01NS035193.

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
