[Supplementary Material]

# Supplementary for: Adabelief optimizer, adapting stepsizes by the belief in observed gradients

## A. Detailed Algorithm of AdaBelief

**Notations**  By the convention in [1], we use the following notations:

- $f(\theta) \in \mathbb{R}, \theta \in \mathbb{R}^d$: $f$ is the loss function to minimize, $\theta$ is the parameter in $\mathbb{R}^d$
- $g_t$: the gradient and step $t$
- $\alpha, \epsilon$: $\alpha$ is the learning rate, default is $10^{-3}$; $\epsilon$ is a small number, typically set as $10^{-8}$
- $\beta_1, \beta_2$: smoothing parameters, typical values are $\beta_1 = 0.9, \beta_2 = 0.999$
- $m_t$: exponential moving average (EMA) of $g_t$
- $v_t, s_t$: $v_t$ is the EMA of $g_t^2$, $s_t$ is the EMA of $(g_t - m_t)^2$
- $\prod_{\mathcal{F},M}(y) = \text{argmin}_{x \in \mathcal{F}} ||M^{1/2}(x - y)||$

---

**Algorithm 1: AdaBelief**

---
**Initialize** $\theta_0$
  $m_0 \leftarrow 0$ , $s_0 \leftarrow 0, t \leftarrow 0$
**While** $\theta_t$ not converged
  $t \leftarrow t + 1$
  $g_t \leftarrow \nabla_\theta f_t(\theta_{t-1})$
  $m_t \leftarrow \beta_1 m_{t-1} + (1 - \beta_1)g_t$
  $s_t \leftarrow \beta_2 s_{t-1} + (1 - \beta_2)(g_t - m_t)^2$
  **If** $AMSGrad$
    $s_t \leftarrow \max(s_t, s_{t-1})$
  **Bias Correction**
    $\widehat{m_t} \leftarrow m_t/(1 - \beta_1^t), \qquad \widehat{s_t} \leftarrow (s_t + \epsilon)/(1 - \beta_2^t)$
  **Update**
    $\theta_t \leftarrow \prod_{\mathcal{F},\sqrt{s_t}}\left(\theta_{t-1} - \widehat{m_t}\frac{\alpha}{\sqrt{\widehat{s_t}}+\epsilon}\right)$

---

## B. Convergence analysis in convex online learning case (Theorem 2.1 in main paper)

For the ease of notation, we absorb $\epsilon$ into $s_t$. Equivalently, $s_t \geq c > 0, \forall t \in [T]$. For simplicity, we omit the debiasing step in theoretical analysis as in [2]. Our analysis can be applied to the de-biased version as well.

**Lemma 0.1.** *[3] For any $Q \in S_+^d$ and convex feasible set $\mathcal{F} \subset \mathbb{R}^d$, suppose $u_1 = \min_{x \in \mathcal{F}} \left|\left|Q^{1/2}(x - z_1)\right|\right|$ and $u_2 = \min_{x \in \mathcal{F}} \left|\left|Q^{1/2}(x - z_2)\right|\right|$, then we have $\left|\left|Q^{1/2}(u_1 - u_2)\right|\right| \leq \left|\left|Q^{1/2}(z_1 - z_2)\right|\right|$.*

**Theorem 0.2.** *Let $\{\theta_t\}$ and $\{s_t\}$ be the sequence obtained by the proposed algorithm, let $0 \leq \beta_2 < 1, \alpha_t = \frac{\alpha}{\sqrt{t}}$, $\beta_{11} = \beta_1$, $0 \leq \beta_{1t} \leq \beta_1 < 1, s_{t-1} \leq s_t, \forall t \in [T]$. Let $\theta \in \mathcal{F}$, where $\mathcal{F} \subset \mathbb{R}^d$ is a convex feasible set with bounded diameter $D_\infty$. Assume $f(\theta)$ is a convex function and $||g_t||_\infty \leq G_\infty/2$ (hence $||g_t - m_t||_\infty \leq G_\infty$) and $s_{t,i} \geq c > 0, \forall t \in [T], \theta \in \mathcal{F}$. Denote the optimal point as $\theta^*$. For $\theta_t$ generated with Algorithm 1, we have the following bound on the regret:*

$$\sum_{t=1}^T f_t(\theta_t) - f_t(\theta^*) \leq \frac{D_\infty^2 \sqrt{T}}{2\alpha(1 - \beta_1)} \sum_{i=1}^d s_{T,i}^{1/2} + \frac{(1+\beta_1)\alpha\sqrt{1 + \log T}}{2\sqrt{c}(1 - \beta_1)^3} \sum_{i=1}^d \left|\left|g_{1:T,i}^2\right|\right|_2$$

$$+ \frac{D_\infty^2}{2(1 - \beta_1)} \sum_{t=1}^T \sum_{i=1}^d \frac{\beta_{1t} s_{t,i}^{1/2}}{\alpha_t}$$

*Proof:*

$$\theta_{t+1} = \prod_{\mathcal{F},\sqrt{s_t}} (\theta_t - \alpha_t s_t^{-1/2} m_t) = \min_{\theta \in \mathcal{F}} \left|\left| s_t^{1/4}[\theta - (\theta_t - \alpha_t s_t^{-1/2} m_t)] \right|\right|$$

Note that $\prod_{\mathcal{F},\sqrt{s_t}}(\theta^*) = \theta^*$ since $\theta^* \in \mathcal{F}$. Use $\theta_i^*$ and $\theta_{t,i}$ to denote the $i$th dimension of $\theta^*$ and $\theta_t$ respectively. From lemma (0.1), using $u_1 = \theta_{t+1}$ and $u_2 = \theta^*$, we have:

$$
\begin{aligned}
\left|\left| s_t^{1/4}(\theta_{t+1} - \theta^*) \right|\right|^2 &\leq \left|\left| s_t^{1/4}(\theta_t - \alpha_t s_t^{-1/2} m_t - \theta^*) \right|\right|^2 \\
&= \left|\left| s_t^{1/4}(\theta_t - \theta^*) \right|\right|^2 + \alpha_t^2 \left|\left| s_t^{-1/4} m_t \right|\right|^2 - 2\alpha_t \langle m_t, \theta_t - \theta^* \rangle \\
&= \left|\left| s_t^{1/4}(\theta_t - \theta^*) \right|\right|^2 + \alpha_t^2 \left|\left| s_t^{-1/4} m_t \right|\right|^2 \\
&\quad - 2\alpha_t \langle \beta_{1t} m_{t-1} + (1 - \beta_{1t}) g_t, \theta_t - \theta^* \rangle
\end{aligned}
\tag{1}
$$

Note that $\beta_1 \in [0,1)$ and $\beta_2 \in [0,1)$, rearranging inequality (1), we have:

$$
\begin{aligned}
\langle g_t, \theta_t - \theta^* \rangle &\leq \frac{1}{2\alpha_t(1 - \beta_{1t})} \left[ \left|\left| s_t^{1/4}(\theta_t - \theta^*) \right|\right|^2 - \left|\left| s_t^{1/4}(\theta_{t+1} - \theta^*) \right|\right|^2 \right] \\
&\quad + \frac{\alpha_t}{2(1 - \beta_{1t})} \left|\left| s_t^{-1/4} m_t \right|\right|^2 - \frac{\beta_{1t}}{1 - \beta_{1t}} \langle m_{t-1}, \theta_t - \theta^* \rangle \\
&\leq \frac{1}{2\alpha_t(1 - \beta_{1t})} \left[ \left|\left| s_t^{1/4}(\theta_t - \theta^*) \right|\right|^2 - \left|\left| s_t^{1/4}(\theta_{t+1} - \theta^*) \right|\right|^2 \right] \\
&\quad + \frac{\alpha_t}{2(1 - \beta_{1t})} \left|\left| s_t^{-1/4} m_t \right|\right|^2 \\
&\quad + \frac{\beta_{1t}}{2(1 - \beta_{1t})} \alpha_t \left|\left| s_t^{-1/4} m_{t-1} \right|\right|^2 + \frac{\beta_{1t}}{2\alpha_t(1 - \beta_{1t})} \left|\left| s_t^{1/4}(\theta_t - \theta^*) \right|\right|^2 \\
&\quad \left( \text{Cauchy-Schwartz and Young's inequality: } ab \leq \frac{a^2\epsilon}{2} + \frac{b^2}{2\epsilon}, \forall \epsilon > 0 \right)
\end{aligned}
\tag{2}
$$

By convexity of $f$, we have:

$$
\begin{aligned}
\sum_{t=1}^{T} f_t(\theta_t) - f_t(\theta^*) &\leq \sum_{t=1}^{T} \langle g_t, \theta_t - \theta^* \rangle \\
&\leq \sum_{t=1}^{T} \Bigg\{ \frac{1}{2\alpha_t(1 - \beta_{1t})} \left[ \left|\left| s_t^{1/4}(\theta_t - \theta^*) \right|\right|^2 - \left|\left| s_t^{1/4}(\theta_{t+1} - \theta^*) \right|\right|^2 \right] \\
&\quad + \frac{1}{2(1 - \beta_{1t})} \alpha_t \left|\left| s_t^{-1/4} m_t \right|\right|^2 + \frac{\beta_{1t}}{2(1 - \beta_{1t})} \alpha_t \left|\left| s_t^{-1/4} m_{t-1} \right|\right|^2 \\
&\quad + \frac{\beta_{1t}}{2\alpha_t(1 - \beta_{1t})} \left|\left| s_t^{1/4}(\theta_t - \theta^*) \right|\right|^2 \Bigg\} \\
&\quad \left( \text{By formula } (2) \right) \\
&\leq \frac{1}{2(1 - \beta_1)} \frac{\left|\left| s_1^{1/4}(\theta_1 - \theta^*) \right|\right|^2}{\alpha_1} \\
&\quad + \frac{1}{2(1 - \beta_1)} \sum_{t=2}^{T} \left[ \frac{\left|\left| s_t^{1/4}(\theta_t - \theta^*) \right|\right|^2}{\alpha_t} - \frac{\left|\left| s_{t-1}^{1/4}(\theta_t - \theta^*) \right|\right|^2}{\alpha_{t-1}} \right] \\
&\quad + \sum_{t=1}^{T} \left[ \frac{1}{2(1 - \beta_1)} \alpha_t \left|\left| s_t^{-1/4} m_t \right|\right|^2 \right] + \sum_{t=2}^{T} \left[ \frac{\beta_1}{2(1 - \beta_1)} \alpha_{t-1} \left|\left| s_{t-1}^{-1/4} m_{t-1} \right|\right|^2 \right] \\
&\quad + \sum_{t=1}^{T} \frac{\beta_{1t}}{2\alpha_t(1 - \beta_{1t})} \left|\left| s_t^{1/4}(\theta_t - \theta^*) \right|\right|^2
\end{aligned}
$$

$$\left(0 \le s_{t-1} \le s_t, 0 \le \alpha_t \le \alpha_{t-1}, 0 \le \beta_{1t} \le \beta_1 < 1\right)$$

$$\le \frac{1}{2(1-\beta_1)} \frac{\left\|s_1^{1/4}(\theta_1 - \theta^*)\right\|^2}{\alpha_1} + \frac{1}{2(1-\beta_1)} \sum_{t=2}^{T} \left\|\theta_t - \theta^*\right\|^2 \left[\frac{s_t^{1/2}}{\alpha_t} - \frac{s_{t-1}^{1/2}}{\alpha_{t-1}}\right]$$

$$+ \frac{1+\beta_1}{2(1-\beta_1)} \sum_{t=1}^{T} \alpha_t \left\|s_t^{-1/4} m_t\right\|^2$$

$$+ \sum_{t=1}^{T} \frac{\beta_{1t}}{2\alpha_t(1-\beta_{1t})} \left\|s_t^{1/4}(\theta_t - \theta^*)\right\|^2$$

$$\le \frac{1}{2(1-\beta_1)} \frac{\left\|s_1^{1/4}(\theta_1 - \theta^*)\right\|^2}{\alpha_1} + \frac{1}{2(1-\beta_1)} \sum_{t=2}^{T} \left\|\theta_t - \theta^*\right\|^2 \left[\frac{s_t^{1/2}}{\alpha_t} - \frac{s_{t-1}^{1/2}}{\alpha_{t-1}}\right]$$

$$+ \frac{1+\beta_1}{2(1-\beta_1)} \sum_{t=1}^{T} \alpha_t \left\|s_t^{-1/4} m_t\right\|^2$$

$$+ \frac{1}{2(1-\beta_1)} \sum_{t=1}^{T} \frac{\beta_{1t}}{\alpha_t} \left\|s_t^{1/4}(\theta_t - \theta^*)\right\|^2$$

$$\left(\text{since } 0 \le \beta_{1t} \le \beta_1 < 1\right) \tag{3}$$

Now bound $\sum_{t=1}^{T} \alpha_t ||s_t^{-1/4} m_t||^2$ in Formula (3), assuming $0 < c \le s_t, \forall t \in [T]$.

$$\sum_{t=1}^{T} \alpha_t \left\|s_t^{-1/4} m_t\right\|^2 = \sum_{t=1}^{T-1} \alpha_t \left\|s_t^{-1/4} m_t\right\|^2 + \alpha_T \left\|s_T^{-1/4} m_T\right\|^2$$

$$\le \sum_{t=1}^{T-1} \alpha_t \left\|s_t^{-1/4} m_t\right\|^2 + \frac{\alpha_T}{\sqrt{c}} \left\|m_T\right\|^2$$

$$= \sum_{t=1}^{T-1} \alpha_t \left\|s_t^{-1/4} m_t\right\|^2 + \frac{\alpha}{\sqrt{cT}} \sum_{i=1}^{d} \left(\sum_{j=1}^{T}(1-\beta_{1,j}) g_{j,i} \prod_{k=1}^{T-j} \beta_{1,T-k+1}\right)^2$$

$$\left(\text{since } m_T = \sum_{j=1}^{T}(1-\beta_{1,j}) g_{j,i} \prod_{k=1}^{T-j} \beta_{1,T-k+1}\right)$$

$$\le \sum_{t=1}^{T-1} \alpha_t \left\|s_t^{-1/4} m_t\right\|^2 + \frac{\alpha}{\sqrt{cT}} \sum_{i=1}^{d} \left(\sum_{j=1}^{T} g_{j,i} \prod_{k=1}^{T-j} \beta_1\right)^2$$

$$(\text{since } 0 < \beta_{1,j} \le \beta_1 < 1)$$

$$= \sum_{t=1}^{T-1} \alpha_t \left\|s_t^{-1/4} m_t\right\|^2 + \frac{\alpha}{\sqrt{cT}} \sum_{i=1}^{d} \left(\sum_{j=1}^{T} \beta_1^{T-j} g_{j,i}\right)^2$$

$$\le \sum_{t=1}^{T-1} \alpha_t \left\|s_t^{-1/4} m_t\right\|^2 + \frac{\alpha}{\sqrt{cT}} \sum_{i=1}^{d} \left(\sum_{j=1}^{T} \beta_1^{T-j}\right)\left(\sum_{j=1}^{T} \beta_1^{T-j} g_{j,i}^2\right)$$

$$\left(Cauchy - Schwartz, \langle u, v\rangle^2 \le \left\|u\right\|^2 \left\|v\right\|^2, u_j = \sqrt{\beta_1^{T-j}}, v_j = \sqrt{\beta_1^{T-j}} g_{j,i}\right)$$

$$= \sum_{t=1}^{T-1} \alpha_t \left\|s_t^{-1/4} m_t\right\|^2 + \frac{\alpha}{\sqrt{cT}} \sum_{i=1}^{d} \frac{1-\beta_1^T}{1-\beta_1} \sum_{j=1}^{T} \beta_1^{T-j} g_{j,i}^2$$

$$\le \sum_{t=1}^{T-1} \alpha_t \left\|s_t^{-1/4} m_t\right\|^2 + \frac{\alpha}{\sqrt{c}(1-\beta_1)} \sum_{i=1}^{d}\sum_{j=1}^{T} \beta_1^{T-j} g_{j,i}^2 \frac{1}{\sqrt{T}}$$

$$\left(\textit{since } 1 - \beta_1^T < 1\right)$$

$$\leq \frac{\alpha}{\sqrt{c}(1-\beta_1)} \sum_{i=1}^{d} \sum_{t=1}^{T} \sum_{j=1}^{t} \beta_1^{t-j} g_{j,i}^2 \frac{1}{\sqrt{t}}$$

$$\left(\textit{Recursively bound each term in the sum } \sum_{t=1}^{T} *\right)$$

$$= \frac{\alpha}{\sqrt{c}(1-\beta_1)} \sum_{i=1}^{d} \sum_{t=1}^{T} g_{t,i}^2 \sum_{j=t}^{T} \frac{\beta_1^{j-t}}{\sqrt{j}}$$

$$\leq \frac{\alpha}{\sqrt{c}(1-\beta_1)} \sum_{i=1}^{d} \sum_{t=1}^{T} g_{t,i}^2 \sum_{j=t}^{T} \frac{\beta_1^{j-t}}{\sqrt{t}}$$

$$\leq \frac{\alpha}{\sqrt{c}(1-\beta_1)^2} \sum_{i=1}^{d} \sum_{t=1}^{T} g_{t,i}^2 \frac{1}{\sqrt{t}}$$

$$\left(\textit{since } \sum_{j=t}^{T} \beta_1^{j-t} = \sum_{j=0}^{T-t} \beta_1^j = \frac{1-\beta_1^{T-t+1}}{1-\beta_1} \leq \frac{1}{1-\beta_1}\right)$$

$$\leq \frac{\alpha}{\sqrt{c}(1-\beta_1)^2} \sum_{i=1}^{d} \left\|g_{1:T,i}^2\right\|_2 \sqrt{\sum_{t=1}^{T} \frac{1}{t}}$$

$$\left(Cauchy - Schwartz, \langle u, v \rangle \leq \|u\|\|v\|, u_t = g_{t,i}^2, v_t = \frac{1}{\sqrt{t}}\right)$$

$$\leq \frac{\alpha\sqrt{1+\log T}}{\sqrt{c}(1-\beta_1)^2} \sum_{i=1}^{d} \left\|g_{1:T,i}^2\right\|_2 \quad \left(\textit{since } \sum_{t=1}^{T} \frac{1}{t} \leq 1 + \log T\right) \tag{4}$$

Apply formula (4) to (3), we have:

$$\sum_{t=1}^{T} f_t(\theta_t) - f_t(\theta^*) \leq \frac{1}{2(1-\beta_1)} \frac{\left\|s_1^{1/4}(\theta_1 - \theta^*)\right\|^2}{\alpha_1} + \frac{1}{2(1-\beta_1)} \sum_{t=2}^{T} \left\|\theta_t - \theta^*\right\|^2 \left[\frac{s_t^{1/2}}{\alpha_t} - \frac{s_{t-1}^{1/2}}{\alpha_{t-1}}\right]$$

$$+ \frac{1+\beta_1}{2(1-\beta_1)} \sum_{t=1}^{T} \alpha_t \left\|s_t^{-1/4} m_t\right\|^2$$

$$+ \frac{1}{2(1-\beta_1)} \sum_{t=1}^{T} \frac{\beta_{1t}}{\alpha_t} \left\|s_t^{1/4}(\theta_t - \theta^*)\right\|^2$$

$$\leq \frac{1}{2(1-\beta_1)} \frac{\left\|s_1^{1/4}(\theta_1 - \theta^*)\right\|^2}{\alpha_1} + \frac{1}{2(1-\beta_1)} \sum_{t=2}^{T} \left\|\theta_t - \theta^*\right\|^2 \left[\frac{s_t^{1/2}}{\alpha_t} - \frac{s_{t-1}^{1/2}}{\alpha_{t-1}}\right]$$

$$+ \frac{(1+\beta_1)\alpha\sqrt{1+\log T}}{2\sqrt{c}(1-\beta_1)^3} \sum_{i=1}^{d} \left\|g_{1:T,i}^2\right\|_2$$

$$+ \frac{1}{2(1-\beta_1)} \sum_{t=1}^{T} \frac{\beta_{1t}}{\alpha_t} \left\|s_t^{1/4}(\theta_t - \theta^*)\right\|^2$$

$$\left(\textit{By formula } (4)\right)$$

$$\leq \frac{1}{2(1-\beta_1)} \sum_{i=1}^{d} \frac{s_{1,i}^{1/2} D_\infty^2}{\alpha_1} + \frac{1}{2(1-\beta_1)} \sum_{t=2}^{T} \sum_{i=1}^{d} D_\infty^2 \left[\frac{s_{t,i}^{1/2}}{\alpha_t} - \frac{s_{t-1,i}^{1/2}}{\alpha_{t-1}}\right]$$

$$+ \frac{(1+\beta_1)\alpha\sqrt{1+\log T}}{2\sqrt{c}(1-\beta_1)^3} \sum_{i=1}^{d} \left\|g_{1:T,i}^2\right\|_2$$

$$+ \frac{D_\infty^2}{2(1-\beta_1)} \sum_{t=1}^{T} \sum_{i=1}^{d} \frac{\beta_{1t} s_{t,i}^{1/2}}{\alpha_t}$$

$$\left(since \ x \in \mathcal{F}, with \ bounded \ diameter \ D_\infty, and \ \frac{s_{t,i}^{1/2}}{\alpha_t} \geq \frac{s_{t-1,i}^{1/2}}{\alpha_{t-1}} \ by \ assumption.\right)$$

$$\leq \frac{D_\infty^2 \sqrt{T}}{2\alpha(1-\beta_1)} \sum_{i=1}^{d} s_{T,i}^{1/2} + \frac{(1+\beta_1)\alpha\sqrt{1+\log T}}{2\sqrt{c}(1-\beta_1)^3} \sum_{i=1}^{d} \left\|g_{1:T,i}^2\right\|_2$$

$$+ \frac{D_\infty^2}{2(1-\beta_1)} \sum_{t=1}^{T} \sum_{i=1}^{d} \frac{\beta_{1t} s_{t,i}^{1/2}}{\alpha_t}$$

$$\left(\alpha_t \geq \alpha_{t+1} \ and \ perform \ telescope \ sum\right) \tag{5}$$

∎

**Corollary 0.2.1.** *Suppose* $\beta_{1,t} = \beta_1 \lambda^t, \ 0 < \lambda < 1$ *in Theorem* (0.2)*, then we have:*

$$\sum_{t=1}^{T} f_t(\theta_t) - f_t(\theta^*) \leq \frac{D_\infty^2 \sqrt{T}}{2\alpha(1-\beta_1)} \sum_{i=1}^{d} s_{T,i}^{1/2} + \frac{(1+\beta_1)\alpha\sqrt{1+\log T}}{2\sqrt{c}(1-\beta_1)^3} \sum_{i=1}^{d} \left\|g_{1:T,i}^2\right\|_2$$

$$+ \frac{D_\infty^2 \beta_1 G_\infty}{2(1-\beta_1)(1-\lambda)^2 \alpha} \tag{6}$$

*Proof:* By sum of arithmetico-geometric series, we have:

$$\sum_{t=1}^{T} \lambda^{t-1} \sqrt{t} \leq \sum_{t=1}^{T} \lambda^{t-1} t \leq \frac{1}{(1-\lambda)^2} \tag{7}$$

Plugging (7) into (5), we can derive the results above. ∎

## C. Convergence analysis for non-convex stochastic optimization (Theorem 2.2 in main paper)

**Assumptions**

- A1, $f$ is differentiable and has $L - Lipschitz$ gradient, $||\nabla f(x) - \nabla f(y)|| \leq L||x - y||, \ \forall x, y$. $f$ is also lower bounded.
- A2, at time $t$, the algorithm can access a bounded noisy gradient, the true gradient is also bounded. *i.e.* $||\nabla f(\theta_t)|| \leq H, \ ||g_t|| \leq H, \ \forall t > 1$.
- A3, The noisy gradient is unbiased, and has independent noise. *i.e.* $g_t = \nabla f(\theta_t) + \zeta_t, \mathbb{E}\zeta_t = 0, \zeta_t \perp \zeta_j, \ \forall j, t \in \mathbb{N}, t \neq j$

**Theorem 0.3.** *[4] Suppose assumptions A1-A3 are satisfied,* $\beta_{1,t}$ *is chosen such that* $0 \leq \beta_{1,t+1} \leq \beta_{1,t} < 1, 0 < \beta_2 < 1, \forall t > 0$. *For some constant* $G$, $\left\|\alpha_t \frac{m_t}{\sqrt{s_t}}\right\| \leq G, \forall t$. *Then Adam-type algorithms yield*

$$\mathbb{E}\left[\sum_{t=1}^{T} \alpha_t \langle \nabla f(\theta_t), \nabla f(\theta_t)/\sqrt{s_t}\rangle\right] \leq$$

$$\mathbb{E}\left[C_1 \sum_{t=1}^{T} \left\|\alpha_t g_t/\sqrt{s_t}\right\|^2 + C_2 \sum_{t=1}^{T} \left\|\frac{\alpha_t}{\sqrt{s_t}} - \frac{\alpha_{t-1}}{\sqrt{s_{t-1}}}\right\|_1 + C_3 \sum_{t=1}^{T} \left\|\frac{\alpha_t}{\sqrt{s_t}} - \frac{\alpha_{t-1}}{\sqrt{s_{t-1}}}\right\|^2\right] + C_4 \tag{8}$$

*where* $C_1, C_2, C_3$ *are constants independent of* $d$ *and* $T$, $C_4$ *is a constant independent of* $T$*, the expectation is taken w.r.t all randomness corresponding to* $\{g_t\}$.

*Furthermore, let* $\gamma_t := \min_{j \in [d]} \min_{\{g_i\}_{i=1}^t} \alpha_i/(\sqrt{s_i})_j$ *denote the minimum possible value of effective stepsize at time* $t$ *over all possible coordinate and past gradients* $\{g_i\}_{i=1}^t$. *The convergence rate of Adam-type algorithm is given by*

$$\min_{t \in [T]} \mathbb{E}\left[\left\|\nabla f(\theta_t)\right\|^2\right] = O\left(\frac{s_1(T)}{s_2(T)}\right) \tag{9}$$

*where* $s_1(T)$ *is defined through the upper bound of RHS of* (8), *and* $\sum_{t=1}^{T} \gamma_t = \Omega(s_2(T))$

**Proof:** We provide the proof from [4] in next section for completeness. ∎

**Theorem 0.4.** *Assume* $\min_{j \in [d]}(s_1)_j \geq c > 0$, *noise in gradient has bounded variance,* $\mathrm{Var}(g_t) = \sigma_t^2 \leq \sigma^2, \forall t \in \mathbb{N}$, *then the AdaBelief algorithm satisfies:*

$$
\begin{aligned}
\min_{t \in [T]} \mathbb{E}\left\|\nabla f(\theta_t)\right\|^2 &\leq \frac{H}{\sqrt{T}\alpha}\left[\frac{C_1\alpha^2(H^2+\sigma^2)(1+\log T)}{c} + C_2\frac{d\alpha}{\sqrt{c}} + C_3\frac{d\alpha^2}{c} + C_4\right] \\
&= \frac{1}{\sqrt{T}}(Q_1 + Q_2\log T)
\end{aligned}
$$

*where*

$$
\begin{aligned}
Q_1 &= \frac{H}{\alpha}\left[\frac{C_1\alpha^2(H^2+\sigma^2)}{c} + C_2\frac{d\alpha}{\sqrt{c}} + C_3\frac{d\alpha^2}{c} + C_4\right] \\
Q_2 &= \frac{HC_1\alpha(H^2+\sigma^2)}{c}
\end{aligned}
$$

**Proof:** We first derive an upper bound of the RHS of formula (8), then derive a lower bound of the LHS of (8).

$$
\begin{aligned}
\mathbb{E}\Big[\sum_{t=1}^{T}\left\|\alpha_t g_t/\sqrt{s_t}\right\|^2\Big] &\leq \frac{1}{c}\mathbb{E}\Big[\sum_{t=1}^{T}\sum_{i=1}^{d}(\alpha_{t,i}g_{t,i})^2\Big] \quad \Big(since\ 0 < c \leq s_t, \forall t \in [T]\Big) \\
&= \frac{1}{c}\sum_{i=1}^{d}\sum_{t=1}^{T}\alpha_t^2\mathbb{E}(g_{t,i})^2 \\
&= \frac{1}{c}\sum_{t=1}^{T}\alpha_t^2\mathbb{E}\left[\left\|\nabla f(\theta_t)\right\|^2 + \left\|\sigma_t\right\|^2\right] \tag{10}
\end{aligned}
$$

$$
\begin{aligned}
\mathbb{E}\Big[\sum_{t=1}^{T}\left\|\frac{\alpha_t}{\sqrt{s_t}} - \frac{\alpha_{t-1}}{\sqrt{s_{t-1}}}\right\|_1\Big] &= \mathbb{E}\Big[\sum_{i=1}^{d}\sum_{t=1}^{T}\frac{\alpha_{t-1}}{\sqrt{s_{t-1,i}}} - \frac{\alpha_t}{\sqrt{s_{t,i}}}\Big] \\
&\qquad\Big(since\ \alpha_t \leq \alpha_{t-1}, s_{t,i} \geq s_{t-1,i}\Big) \\
&= \mathbb{E}\Big[\sum_{i=1}^{d}\frac{\alpha_1}{\sqrt{s_{1,i}}} - \frac{\alpha_T}{\sqrt{s_{T,i}}}\Big] \\
&\leq \mathbb{E}\Big[\sum_{i=1}^{d}\frac{\alpha_1}{\sqrt{s_{1,i}}}\Big] \\
&\leq \frac{d\alpha}{\sqrt{c}} \quad \Big(since\ 0 < c \leq s_t, 0 \leq \alpha_t \leq \alpha_1 = \alpha, \forall t\Big) \tag{11}
\end{aligned}
$$

$$
\mathbb{E}\Big[\sum_{t=1}^{T}\left\|\frac{\alpha_t}{\sqrt{s_t}} - \frac{\alpha_{t-1}}{\sqrt{s_{t-1}}}\right\|^2\Big] = \mathbb{E}\Big[\sum_{t=1}^{T}\sum_{i=1}^{d}\Big(\frac{\alpha_t}{\sqrt{s_t}} - \frac{\alpha_{t-1}}{\sqrt{s_{t-1}}}\Big)_i^2\Big]
$$

$$\leq \mathbb{E}\Big[\sum_{t=1}^{T}\sum_{i=1}^{d}\Big|\frac{\alpha_t}{\sqrt{s_t}} - \frac{\alpha_{t-1}}{\sqrt{s_{t-1}}}\Big|_i \frac{\alpha}{\sqrt{c}}\Big]$$

$$\Big(Since \ \Big|\frac{\alpha_t}{\sqrt{s_t}} - \frac{\alpha_{t-1}}{\sqrt{s_{t-1}}}\Big| = \frac{\alpha_{t-1}}{\sqrt{s_{t-1}}} - \frac{\alpha_t}{\sqrt{s_t}} \leq \frac{\alpha_{t-1}}{\sqrt{s_{t-1}}} \leq \frac{\alpha}{\sqrt{c}}\Big)$$

$$\leq \frac{d\alpha^2}{c} \ \Big(By \ (11)\Big) \tag{12}$$

Next we derive the lower bound of LHS of (8).

$$\mathbb{E}\Big[\sum_{t=1}^{T}\alpha_t\langle\nabla f(\theta_t), \frac{\nabla f(\theta_t)}{\sqrt{s_t}}\rangle\Big] \geq \frac{1}{H}\mathbb{E}\Big[\sum_{t=1}^{T}\alpha_t\big|\big|\nabla f(\theta_t)\big|\big|^2\Big] \geq \frac{\alpha\sqrt{T}}{H}\min_{t\in[T]}\mathbb{E}\big|\big|\nabla f(\theta_t)\big|\big|^2 \tag{13}$$

Combining (10), (11), (12) and (13) to (8), we have:

$$\frac{\alpha\sqrt{T}}{H}\min_{t\in[T]}\mathbb{E}\big|\big|\nabla f(\theta_t)\big|\big|^2 \leq \mathbb{E}\Big[\sum_{t=1}^{T}\alpha_t\langle\nabla f(\theta_t), \frac{\nabla f(\theta_t)}{\sqrt{s_t}}\rangle\Big]$$

$$\leq \mathbb{E}\Big[C_1\sum_{t=1}^{T}\big|\big|\alpha_t g_t/\sqrt{s_t}\big|\big|^2 + C_2\sum_{t=1}^{T}\big|\big|\frac{\alpha_t}{\sqrt{s_t}} - \frac{\alpha_{t-1}}{\sqrt{s_{t-1}}}\big|\big|_1 + C_3\sum_{t=1}^{T}\big|\big|\frac{\alpha_t}{\sqrt{s_t}} - \frac{\alpha_{t-1}}{\sqrt{s_{t-1}}}\big|\big|^2\Big] + C_4$$

$$\leq \frac{C_1}{c}\sum_{t=1}^{T}\mathbb{E}\Big[\alpha_t^2\big|\big|\nabla f(\theta_t)\big|\big|^2 + \alpha_t^2\big|\big|\sigma_t\big|\big|^2\Big] + C_2\frac{d\alpha}{\sqrt{c}} + C_3\frac{d\alpha^2}{c} + C_4 \tag{14}$$

$$\leq \frac{C_1}{c}\sum_{t=1}^{T}\mathbb{E}\Big[\alpha_t^2(H^2 + \sigma^2)\Big] + C_2\frac{d\alpha}{\sqrt{c}} + C_3\frac{d\alpha^2}{c} + C_4$$

$$\leq \frac{C_1\alpha^2(H^2 + \sigma^2)(1+\log T)}{c} + C_2\frac{d\alpha}{\sqrt{c}} + C_3\frac{d\alpha^2}{c} + C_4 \tag{15}$$

$$\Big(since \ \alpha_t = \frac{\alpha}{\sqrt{t}}, \ \sum_{t=1}^{T}\frac{1}{t} \leq 1+\log T\Big)$$

Re-arranging above inequality, we have

$$\min_{t\in[T]}\mathbb{E}\big|\big|\nabla f(\theta_t)\big|\big|^2 \leq \frac{H}{\sqrt{T}\alpha}\Big[\frac{C_1\alpha^2(H^2+\sigma^2)(1+\log T)}{c} + C_2\frac{d\alpha}{\sqrt{c}} + C_3\frac{d\alpha^2}{c} + C_4\Big]$$

$$= \frac{1}{\sqrt{T}}(Q_1 + Q_2\log T) \tag{16}$$

where

$$Q_1 = \frac{H}{\alpha}\Big[\frac{C_1\alpha^2(H^2+\sigma^2)}{c} + C_2\frac{d\alpha}{\sqrt{c}} + C_3\frac{d\alpha^2}{c} + C_4\Big] \tag{17}$$

$$Q_2 = \frac{HC_1\alpha(H^2+\sigma^2)}{c} \tag{18}$$

∎

**Corollary 0.4.1.** *If $c > C_1 H$ and assumptions for Theorem 0.3 are satisfied, we have:*

$$\frac{1}{T}\sum_{t=1}^{T}\mathbb{E}\Big[\alpha_t^2\big|\big|\nabla f(\theta_t)\big|\big|^2\Big] \leq \frac{1}{T}\frac{1}{\frac{1}{H} - \frac{C_1}{c}}\Big[\frac{C_1\alpha^2\sigma^2}{c}(1+\log T) + C_2\frac{d\alpha}{\sqrt{c}} + C_3\frac{d\alpha^2}{c} + C_4\Big] \tag{19}$$

*Proof:* From (13) and (14), we have

$$\frac{1}{H}\mathbb{E}\Big[\sum_{t=1}^{T}\alpha_t\big|\big|\nabla f(\theta_t)\big|\big|^2\Big] \leq \mathbb{E}\Big[\sum_{t=1}^{T}\alpha_t\langle\nabla f(\theta_t), \frac{\nabla f(\theta_t)}{\sqrt{s_t}}\rangle\Big]$$

$$\leq \frac{C_1}{c} \sum_{t=1}^{T} \mathbb{E}\Big[\alpha_t^2 \big\|\nabla f(\theta_t)\big\|^2 + \alpha_t^2 \big\|\sigma_t\big\|^2\Big] + C_2 \frac{d\alpha}{\sqrt{c}} + C_3 \frac{d\alpha^2}{c} + C_4 \tag{20}$$

By re-arranging, we have

$$\big(\frac{1}{H} - \frac{C_1}{c}\big) \sum_{t=1}^{T} \mathbb{E}\Big[\alpha_t^2 \big\|\nabla f(\theta_t)\big\|^2\Big] \leq \frac{C_1}{c} \sum_{t=1}^{T} \mathbb{E}\Big[\alpha_t^2 \big\|\sigma_t\big\|^2\Big] + C_2 \frac{d\alpha}{\sqrt{c}} + C_3 \frac{d\alpha^2}{c} + C_4$$

$$\leq \frac{C_1 \alpha^2 \sigma^2}{c}\big(1 + \log T\big) + C_2 \frac{d\alpha}{\sqrt{c}} + C_3 \frac{d\alpha^2}{c} + C_4 \tag{21}$$

By assumption, $\frac{1}{H} - \frac{C_1}{c} > 0$, then we have

$$\sum_{t=1}^{T} \mathbb{E}\Big[\alpha_t^2 \big\|\nabla f(\theta_t)\big\|^2\Big] \leq \frac{1}{\frac{1}{H} - \frac{C_1}{c}} \left[ \frac{C_1 \alpha^2 \sigma^2}{c}\big(1 + \log T\big) + C_2 \frac{d\alpha}{\sqrt{c}} + C_3 \frac{d\alpha^2}{c} + C_4 \right] \tag{22}$$

∎

## D. Proof of Theorem 0.3

**Lemma 0.5.** *[4] Let $\theta_0 \triangleq \theta_1$ in the Algorithm, consider the sequence*

$$z_t = \theta_t + \frac{\beta_{1,t}}{1 - \beta_{1,t}}(\theta_t - \theta_{t-1}), \forall t \geq 2$$

*The following holds true:*

$$z_{t+1} - z_t = -\Big(\frac{\beta_{1,t+1}}{1 - \beta_{1,t+1}} - \frac{\beta_{1,t}}{1 - \beta_{1,t}}\Big) \frac{\alpha_t m_t}{\sqrt{s_t}}$$

$$- \frac{\beta_{1,t}}{1 - \beta_{1,t}} \Big(\frac{\alpha_t}{\sqrt{s_t}} - \frac{\alpha_{t-1}}{\sqrt{s_{t-1}}}\Big) m_{t-1} - \frac{\alpha_t g_t}{\sqrt{s_t}}, \forall t > 1 \tag{23}$$

*and*

$$z_2 - z_1 = -\Big(\frac{\beta_{1,2}}{1 - \beta_{1,2}} - \frac{\beta_{1,1}}{1 - \beta_{1,1}}\Big) \frac{\alpha_1 m_1}{\sqrt{v_1}} - \frac{\alpha_1 g_1}{\sqrt{v_1}} \tag{24}$$

**Lemma 0.6.** *[4] Suppose that the conditions in Theorem (0.3) hold, then*

$$\mathbb{E}\Big[f(z_{t+1} - f(z_t))\Big] \leq \sum_{i=1}^{6} T_i \tag{25}$$

*where*

$$T_1 = -\mathbb{E}\Big[\sum_{i=1}^{t} \langle \nabla f(z_i), \frac{\beta_{1,i}}{1 - \beta_{1,i}}\Big(\frac{\alpha_i}{\sqrt{v_i}} - \frac{\alpha_{i-1}}{\sqrt{v_{i-1}}}\Big) m_{i-1} \rangle \Big] \tag{26}$$

$$T_2 = -\mathbb{E}\Big[\sum_{i=1}^{t} \alpha_i \langle \nabla f(z_i), \frac{g_i}{\sqrt{v_i}} \rangle \Big] \tag{27}$$

$$T_3 = -\mathbb{E}\Big[\sum_{i=1}^{t} \langle \nabla f(z_i), \Big(\frac{\beta_{1,i+1}}{1 - \beta_{1,i+1}} - \frac{\beta_i}{1 - \beta_i}\Big) \frac{\alpha_i m_i}{\sqrt{v_i}} \rangle \Big] \tag{28}$$

$$T_4 = \mathbb{E}\Big[\sum_{i=1}^{t} \frac{3L}{2} \Big\|\Big(\frac{\beta_{1,i+1}}{1 - \beta_{1,i+1}} - \frac{\beta_{1,i}}{1 - \beta_{1,i}}\Big) \frac{\alpha_i m_i}{\sqrt{v_i}}\Big\|^2 \Big] \tag{29}$$

$$T_5 = \mathbb{E}\Big[\sum_{i=1}^{t} \frac{3L}{2} \Big\|\frac{\beta_{1,i}}{1 - \beta_{1,i}}\Big(\frac{\alpha_i}{\sqrt{v_i}} - \frac{\alpha_{i-1}}{\sqrt{v_{i-1}}}\Big) m_{i-1}\Big\|^2 \Big] \tag{30}$$

$$T_6 = \mathbb{E}\Big[\sum_{i=1}^{t} \frac{3L}{2} \Big\|\frac{\alpha_i g_i}{\sqrt{v_i}}\Big\|^2 \Big] \tag{31}$$

**Lemma 0.7.** *[4] Suppose that the condition in Theorem 0.3 hold, $T_1$ in (26) can be bounded as:*

$$T_1 = -\mathbb{E}\Big[\sum_{i=1}^{t}\langle\nabla f(z_i), \frac{\beta_{1,i}}{1-\beta_{1,i}}\Big(\frac{\alpha_i}{\sqrt{v_i}} - \frac{\alpha_{i-1}}{\sqrt{v_{i-1}}}\Big)m_{i-1}\rangle\Big]$$

$$\leq H^2\frac{\beta_1}{1-\beta_1}\mathbb{E}\Big[\sum_{i=2}^{t}\sum_{j=1}^{d}\Big|\Big(\frac{\alpha_i}{\sqrt{v_i}} - \frac{\alpha_{i-1}}{\sqrt{v_{i-1}}}\Big)_j\Big|\Big] \tag{32}$$

**Lemma 0.8.** *[4] Suppose the conditions in Theorem 0.3 are satisfied, then $T_3$ in (28) can be bounded as*

$$T_3 = -\mathbb{E}\Big[\sum_{i=1}^{t}\langle\nabla f(z_i), \Big(\frac{\beta_{1,i+1}}{1-\beta_{1,i+1}} - \frac{\beta_i}{1-\beta_i}\Big)\frac{\alpha_i m_i}{\sqrt{v_i}}\rangle\Big]$$

$$\leq \Big(\frac{\beta_1}{1-\beta_1} - \frac{\beta_{1,t+1}}{1-\beta_{1,t+1}}\Big)(H^2 + G^2) \tag{33}$$

**Lemma 0.9.** *[4] Suppose assumptions in Theorem 0.3 are satisfied, then $T_4$ in (29) can be bounded as:*

$$T_4 = \mathbb{E}\Big[\sum_{i=1}^{t}\frac{3L}{2}\Big\|\Big(\frac{\beta_{1,i+1}}{1-\beta_{1,i+1}} - \frac{\beta_{1,i}}{1-\beta_{1,i}}\Big)\frac{\alpha_i m_i}{\sqrt{v_i}}\Big\|^2\Big]$$

$$\leq \frac{3L}{2}\Big(\frac{\beta_1}{1-\beta_1} - \frac{\beta_{1,t+1}}{1-\beta_{1,t+1}}\Big)^2 G^2 \tag{34}$$

**Lemma 0.10.** *[4] Suppose the assumptions in Theorem 0.3 are satisfied, then $T_5$ in (30) can be bounded as:*

$$T_5 = \mathbb{E}\Big[\sum_{i=1}^{t}\frac{3L}{2}\Big\|\frac{\beta_{1,i}}{1-\beta_{1,i}}\Big(\frac{\alpha_i}{\sqrt{v_i}} - \frac{\alpha_{i-1}}{\sqrt{v_{i-1}}}\Big)m_{i-1}\Big\|^2\Big]$$

$$\leq \frac{3L}{2}\Big(\frac{\beta_1}{1-\beta_1}\Big)^2 H^2\mathbb{E}\Big[\sum_{i=2}^{t}\sum_{j=1}^{d}\Big(\frac{\alpha_i}{\sqrt{v_i}} - \frac{\alpha_{i-1}}{\sqrt{v_{i-1}}}\Big)_j^2\Big] \tag{35}$$

**Lemma 0.11.** *[4] Suppose the assumptions in Theorem 8 are satisfied, then $T_2$ in (27) are bounded as:*

$$T_2 = -\mathbb{E}\Big[\sum_{i=1}^{t}\alpha_i\langle\nabla f(z_i), \frac{g_i}{\sqrt{v_i}}\rangle\Big]$$

$$\leq \mathbb{E}\sum_{i=2}^{t}\frac{1}{2}\Big\|\frac{\alpha_i g_i}{\sqrt{v_i}}\Big\|^2 + L^2\Big(\frac{\beta_1}{1-\beta_1}\Big)^2\Big(\frac{1}{1-\beta_1}\Big)^2\mathbb{E}\Big[\sum_{j=1}^{d}\sum_{i=2}^{t-1}\Big(\frac{\alpha_i g_i}{\sqrt{v_i}}\Big)_j^2\Big]$$

$$+ L^2 H^2\Big(\frac{\beta_1}{1-\beta_1}\Big)^4\Big(\frac{1}{1-\beta_1}\Big)^2\mathbb{E}\Big[\sum_{j=1}^{d}\sum_{i=2}^{t-1}\Big(\frac{\alpha_i}{\sqrt{v_i}} - \frac{\alpha_{i-1}}{\sqrt{v_{i-1}}}\Big)_j^2\Big]$$

$$+ 2H^2\mathbb{E}\Big[\sum_{j=1}^{d}\sum_{i=2}^{t}\Big|\Big(\frac{\alpha_i}{\sqrt{v_i}} - \frac{\alpha_{i-1}}{\sqrt{v_{i-1}}}\Big)_j\Big|\Big]$$

$$+ 2H^2\mathbb{E}\Big[\sum_{j=1}^{d}\Big(\frac{\alpha_1}{\sqrt{v_1}}\Big)_j\Big]$$

$$- \mathbb{E}\Big[\sum_{i=1}^{t}\alpha_i\langle\nabla f(x_i), \nabla f(x_i)/\sqrt{v_i}\rangle\Big] \tag{36}$$

**Proof of Theorem 0.3**

We provide the proof from [4] for completeness. We combine Lemma 0.5, 0.6, 0.7, 0.8, 0.9, 0.10 and 0.11 to bound the objective.

$$\mathbb{E}\Big[f(z_{t+1}) - f(z_t)\Big] \leq \sum_{i=1}^{6} T_i$$

$$\leq H^2 \frac{\beta_1}{1-\beta_1} \mathbb{E}\left[\sum_{i=2}^{t}\sum_{j=1}^{d}\Big|\Big(\frac{\alpha_i}{\sqrt{v_i}} - \frac{\alpha_{i-1}}{\sqrt{v_{i-1}}}\Big)_j\Big|\right]$$

$$+ \Big(\frac{\beta_1}{1-\beta_1} - \frac{\beta_{1,t+1}}{1-\beta_{1,t+1}}\Big)(H^2 + G^2)$$

$$+ \frac{3L}{2}\Big(\frac{\beta_1}{1-\beta_1} - \frac{\beta_{1,t}}{1-\beta_{1,t}}\Big)^2 G^2$$

$$+ \frac{3L}{2}\Big(\frac{\beta_1}{1-\beta_1}\Big)^2 H^2 \mathbb{E}\left[\sum_{i=2}^{t}\sum_{j=1}^{d}\Big(\frac{\alpha_i}{\sqrt{v_i}} - \frac{\alpha_{i-1}}{\sqrt{v_{i-1}}}\Big)_j^2\right]$$

$$+ \mathbb{E}\sum_{i=2}^{t}\frac{1}{2}\Big\|\frac{\alpha_i g_i}{\sqrt{v_i}}\Big\|^2 + L^2\Big(\frac{\beta_1}{1-\beta_1}\Big)^2\Big(\frac{1}{1-\beta_1}\Big)^2 \mathbb{E}\left[\sum_{j=1}^{d}\sum_{i=2}^{t-1}\Big(\frac{\alpha_i g_i}{\sqrt{v_i}}\Big)_j^2\right]$$

$$+ L^2 H^2\Big(\frac{\beta_1}{1-\beta_1}\Big)^4\Big(\frac{1}{1-\beta_1}\Big)^2 \mathbb{E}\left[\sum_{j=1}^{d}\sum_{i=2}^{t-1}\Big(\frac{\alpha_i}{\sqrt{v_i}} - \frac{\alpha_{i-1}}{\sqrt{v_{i-1}}}\Big)_j^2\right]$$

$$+ 2H^2 \mathbb{E}\left[\sum_{j=1}^{d}\sum_{i=2}^{t}\Big|\Big(\frac{\alpha_i}{\sqrt{v_i}} - \frac{\alpha_{i-1}}{\sqrt{v_{i-1}}}\Big)_j\Big|\right]$$

$$+ 2H^2 \mathbb{E}\left[\sum_{j=1}^{d}\Big(\frac{\alpha_1}{\sqrt{v_1}}\Big)_j\right]$$

$$- \mathbb{E}\left[\sum_{i=1}^{t}\alpha_i\langle\nabla f(x_i), \nabla f(x_i)/\sqrt{v_i}\rangle\right]$$

$$\leq \mathbb{E}\Big[C_1\sum_{t=1}^{T}\Big\|\alpha_t g_t/\sqrt{s_t}\Big\|^2 + C_2\sum_{t=1}^{T}\Big\|\frac{\alpha_t}{\sqrt{s_t}} - \frac{\alpha_{t-1}}{\sqrt{s_{t-1}}}\Big\|_1$$

$$+ C_3\sum_{t=1}^{T}\Big\|\frac{\alpha_t}{\sqrt{s_t}} - \frac{\alpha_{t-1}}{\sqrt{s_{t-1}}}\Big\|^2\Big] + C_4 \tag{37}$$

The constants are defined below:

$$C_1 \triangleq \frac{3}{2}L + \frac{1}{2} + L^2\frac{\beta_1}{1-\beta_1}\Big(\frac{1}{1-\beta_1}\Big)^2 \tag{38}$$

$$C_2 \triangleq H^2\frac{\beta_1}{1-\beta_1} + 2H^2 \tag{39}$$

$$C_3 \triangleq \Big[1 + L^2\Big(\frac{1}{1-\beta_1}\Big)^2\Big(\frac{\beta_1}{1-\beta_1}\Big)\Big]H^2\Big(\frac{\beta_1}{1-\beta_1}\Big)^2 \tag{40}$$

$$C_4 \triangleq \Big(\frac{\beta_1}{1-\beta_1}\Big)(H^2 + G^2) + \Big(\frac{\beta_1}{1-\beta_1}\Big)^2 G^2 + 2H^2\mathbb{E}\big[\|\alpha_1/\sqrt{v_1}\|_1\big] + \mathbb{E}[f(z_1) - f(z^*)] \tag{41}$$

■

# E. Bayesian interpretation of AdaBelief

We analyze AdaBelief from a Bayesian perspective.

**Theorem 0.12.** *Assume the gradient follows a Gaussian prior with uniform diagonal covariance,* $\tilde{g} \sim \mathcal{N}(0, \sigma^2 I)$; *assume the observed gradient follows a Gaussian distribution,* $g \sim \mathcal{N}(\tilde{g}, C)$, *where $C$ is some covariance matrix. Then the posterior is:* $\tilde{g}\big|g, C \sim \mathcal{N}\left((I + \frac{C}{\sigma^2})^{-1}g, (\frac{I}{\sigma^2} + C^{-1})^{-1}\right)$

We skip the proof, which is a direct application of the Bayes rule in the Gaussian distribution case as in [5]. If $g$ is averaged across a batch of size $n$, we can replace $C$ with $\frac{C}{n}$.

According to Theorem 0.12, the gradient descent direction with maximum expected gain is:

$$\mathbb{E}\big[\tilde{g}\big|g, C\big] = (I + \frac{C}{\sigma^2})^{-1}g = \sigma^2(\sigma^2 I + C)^{-1}g \propto (\sigma^2 I + C)^{-1}g \tag{42}$$

Denote $\epsilon = \sigma^2$, then adaptive optimizers update in the direction $(\epsilon I + C)^{-1}g$; considering the noise in $g_t$, in practice most optimizers replace $g_t$ with its EMA $m_t$, hence the update direction is $(\epsilon I + C)^{-1}m_t$. In practice, adaptive methods such as Adam and AdaGrad replace $(\epsilon I + C)^{-1/2}(\epsilon I + C)^{-1/2}m_t$ with $\alpha I(\epsilon I + C)^{-1/2}m_t$ for numerical stability, where $\alpha$ is some predefined learning rate. Both Adam and AdaBelief take this form; their difference is in the estimate of $C$: Adam uses an *uncentered* approximation $C_{Adam} \approx \mathrm{EMA}\,\mathrm{diag}(g_t g_t^\top)$, while AdaBelief uses a *centered* approximation $C_{AdaBelief} \approx \mathrm{EMA}\,\mathrm{diag}[(g_t - \mathbb{E}g_t)(g_t - \mathbb{E}g_t)^\top]$. Note that the definition of $C$ is the *covariance* hence it is *centered*. Note that for the $i$th parameter, $\mathbb{E}(g_t^i)^2 = (\mathbb{E}g_t^i)^2 + \mathrm{Var}(g_t^i)$, so when $\mathrm{Var}\,g_t^i \ll \|\mathbb{E}g_t^i\|$, we have $C_{AdaBelief}^i < C_{Adam}^i$, and AdaBelief behaves closer to the ideal and takes a larger step than Adam because $C$ is in the denominator.

From a practical perspective, $\epsilon$ can be interpreted as a numerical term to avoid division by 0; from the Bayesian perspective, $\epsilon$ represents our prior on $g_t$, with a larger $\epsilon$ indicating a larger $\sigma^2$. Note that as the network evolves with training, the distribution of the gradient is distorted (an example with Adam is shown in Fig. 2 of [6]), hence the Gaussian prior might not match the true distribution. To solve the mismatch between prior and the true distribution, it might be reasonable to use a weak prior during late stages of training (e.g., let $\sigma^2$ grow at late training phases, and when $\sigma^2 \to \infty$ reduces to a uniform prior). We only provide a Bayesian perspective here, and leave the detailed discussion to future works.

| Train accuracy ~ Training epoch | Train accuracy ~ Training epoch | Train accuracy ~ Training epoch |
|---|---|---|
| (a) VGG11 on Cifar10 | (b) ResNet34 on Cifar10 | (c) DenseNet121 on Cifar10 |

| Test accuracy ~ Training epoch | Test accuracy ~ Training epoch | Test accuracy ~ Training epoch |
|---|---|---|
| (d) VGG11 on Cifar10 | (e) ResNet34 on Cifar10 | (f) DenseNet121 on Cifar10 |

Figure 1: Training (top row) and test (bottom row) accuracy of CNNs on Cifar10 dataset. We report confidence interval $[\mu \pm \sigma]$ of 3 independent runs.

## F. Experimental Details

### 1. Image classification with CNNs on Cifar

We performed experiments based on the official implementation[1] of AdaBound [7], and exactly replicated the results of AdaBound as reported in [7]. We then experimented with different optimizers under the same setting: for all experiments, the model is trained for 200 epochs with a batch size of 128, and the learning rate is multiplied by 0.1 at epoch 150. We performed extensive hyper-parameter search as described in the main paper. In the main paper we only report test accuracy; here we report both training and test accuracy in Fig. 1 and Fig. 2. AdaBelief not only achieves the highest test accuracy, but also a smaller gap between training and test accuracy compared with other optimizers such as Yogi.

### 2. Image Classification on ImageNet

We experimented with a ResNet18 on ImageNet classication task. For SGD, we use the same learning rate schedule as [8], with an initial learning rate of 0.1, and multiplied by 0.1 at epoch 30 and 60; for AdaBelief, we use an initial learning rate of 0.001, and decayed it at epoch 70 and 80. Weight decay is set as $10^{-4}$ for both cases. To match the settings in [9] and [6], we use decoupled weight decay. As shown in Fig. 3, AdaBelief achieves an accuracy very close to SGD, closing the generalization gap between adaptive methods and SGD. Meanwhile, when trained with a large learning rate (0.1 for SGD, 0.001 for AdaBelief), AdaBelief achieves faster convergence than SGD in the initial phase.

### 3. Robustness to hyperparameters

**Robustness to** $\epsilon$  We test the performances of AdaBelief and Adam with different values of $\epsilon$ varying from $10^{-4}$ to $10^{-9}$ in a log-scale grid. We perform experiments with a ResNet34 on Cifar10 dataset, and summarize the results in Fig. 4. Compared with Adam, AdaBelief is slightly more sensitive to the choice of $\epsilon$, and achieves the highest accuracy at the default valiue $\epsilon = 10^{-8}$;

Figure 2: Training (top row) and test (bottom row) accuracy of CNNs on Cifar10 dataset. We report confidence interval $[\mu \pm \sigma]$ of 3 independent runs.

Figure 3: Training and test accuracy (top-1) of ResNet18 on ImageNet.

AdaBelief achieves accuracy higher than $94\%$ for all $\epsilon$ values, consistently outperforming Adam which achieves an accuracy around $93\%$.

**Robustness to learning rate** We test the performance of AdaBelief with different learning rates. We experiment with a VGG11 network on Cifar10, and display the results in Fig. 5. For a large range of learning rates from $5 \times 10^{-4}$ to $3 \times 10^{-3}$, compared with Adam, AdaBelief generates higher test accuracy curve, and is more robust to the change of learning rate.

## 4. Experiments with LSTM on language modeling

We experiment with LSTM models on Penn-TreeBank dataset, and report the results in Fig. 6. Our experiments are based on this implementation [2]. Results $[\mu \pm \sigma]$ are measured across 3 runs with independent initialization. For completeness, we plot both the training and test curves.

We use the default parameters $\alpha = 0.001, \beta_1 = 0.9, \beta_2 = 0.999, \epsilon = 10^{-8}$ for 2-layer and 3-layer models; for 1-layer model we set $\epsilon = 10^{-12}$ and set other parameters as default. For simple models

Figure 4: Training (top row) and test (bottom row) accuracy of ResNet34 on Cifar10, trained with AdaBelief (left column) and Adam (right column) using different values of $\epsilon$. Note that AdaBelief achieves an accuracy above $94\%$ for all $\epsilon$ values, while Adam's accuracy is consistently below $94\%$.

Figure 5: Training (top row) and test (bottom row) accuracy of VGG on Cifar10, trained with AdaBelief (left column) and Adam (right column) using different values of learning rate.

| (a) 1-layer LSTM | (b) 2-layer LSTM | (c) 3-layer LSTM |

| (d) 1-layer LSTM | (e) 2-layer LSTM | (f) 3-layer LSTM |

Figure 6: Training (top row) and test (bottom row) perplexity on Penn-TreeBank dataset, lower is better.

Table 1: Structure of GAN

| Generator | Discriminator |
|---|---|
| ConvTranspose ([inchannel = 100, outchannel = 512, kernel = 4×4, stride = 1]) | Conv2D([inchannel=3, outchannel=64, kernel = 4×4, stride=2]) |
| BN-ReLU | LeakyReLU |
| ConvTranspose ([inchannel = 512, outchannel = 256, kernel = 4×4, stride = 2]) | Conv2D([inchannel=64, outchannel=128, kernel = 4×4, stride=2]) |
| BN-ReLU | BN-LeakyReLU |
| ConvTranspose ([inchannel = 256, outchannel = 128, kernel = 4×4, stride = 2]) | Conv2D([inchannel=128, outchannel=256, kernel = 4×4, stride=2]) |
| BN-ReLU | BN-LeakyReLU |
| ConvTranspose ([inchannel = 128, outchannel = 64, kernel = 4×4, stride = 2]) | Conv2D([inchannel=256, outchannel=512, kernel = 4×4, stride=2]) |
| BN-ReLU | BN-LeakyReLU |
| ConvTranspose ([inchannel = 64, outchannel = 3, kernel = 4×4, stride = 2]) | Linear(-1, 1) |
| Tanh | |

(1-layer LSTM), AdaBelief's perplexity is very close to other optimizers; on complicated models, AdaBelief achieves a significantly lower perplexity on the test set.

## 5. Experiments with GAN

We experimented with a WGAN [10] and WGAN-GP [11]. The code is based on several public github repositories [3],[4]. We summarize network structure in Table 1. For WGAN, the weight of discriminator is clipped within $[-0.01, 0.01]$; for WGAN-GP, the weight for gradient-penalty is set as 10.0, as recommended by the original implementation. For each optimizer, we perform 5 independent runs. We train the model for 100 epochs, generate 64,000 fake samples (60,000 real images in Cifar10), and measure the Frechet Inception Distance (FID) [12] between generated samples and real samples. Our implementation on FID heavily relies on an open-source implementation[5]. We report the FID scores in the main paper, and demonstrate fake samples in Fig. 7 and Fig. 8 for WGAN and WGAN-GP respectively.

We also experimented with Spectral Normalization GAN based on a public repository [6]. For this experiment, we set $\epsilon = 10^{-16}$ and use the rectification technique as in RAdam. Other hyperparamters and training schemes are the same as in the repository.

(a) AdaBelief      (b) RMSProp      (c) Adam

(d) RAdam      (e) Yogi      (f) Fromage

(g) MSVAG      (h) AdaBound      (i) SGD

Figure 7: Fake samples from WGAN trained with different optimizers.

(a) AdaBelief      (b) RMSProp      (c) Adam

(d) RAdam      (e) Yogi      (f) Fromage

(g) MSVAG      (h) AdaBound      (i) SGD

Figure 8: Fake samples from WGAN-GP trained with different optimizers.

## Footnotes

[1]`https://github.com/Luolc/AdaBound`

[2] https://github.com/salesforce/awd-lstm-lm

[3] https://github.com/pytorch/examples

[4] https://github.com/eriklindernoren/PyTorch-GAN

[5] https://github.com/mseitzer/pytorch-fid

[6] https://github.com/POSTECH-CVLab/PyTorch-StudioGAN