[Reviews · NeurIPS 2020]

Review 1

Summary and Contributions: The paper proposes a modification of Adam where in the second order momentum term is exchanged for a centered (or central) second order momentum. That is, the v_t terms which are estimates of the diagonal of the uncentered covariance matrix, are exchanged for an estimate of the central second order momentum of the gradients. The paper then offers some intuitive justifications and very illustrative figures on toy problems for their modifications. Followed by a very well designed battery of experiments comparing against many (if not almost all) relevant benchmarks. The idea is interesting, and the experiments (with complementary videos) are very well done. On the downside, there are many issues with the convergence theorems, and some unsubstantiated remarks throughout.

Strengths: - The intuitive explanations for the need of a centered second order momentum estimate in and around Figure 1 and Table 1 are compelling. - The paper provides beautiful vizualization of toy 2d/3d examples illustrating the convergence of the new algorithm Adabelief as compared to Adam and SGD. - Very strong experimental results, with a broad set of benchmarks (SGD, AdaBound, Yogi, Adam, MSVAG, RAdam, Fromage and AdamW). Furthermore the parameter tuning for Adabelief was done over either the same grid as competing method or set to default values.

Weaknesses: 1- The paper contains some unsubstantiated claims. For instance: * Line 145: "Although the above cases are simple, they occur frequently in deep learning, hence we expect AdaBelief to outperform Adam in general cases" and line 150 "most networks behave(s) like (the) case (in) Fig. 3(b)." This statement is not substantiated. Although ReLU losses are somewhat similar to L1 loss (both are composed of two linear components), if one consider the composition resulting from several layers of a deep neural net, the resulting loss function is no longer a simple piecewise linear convex function such as the examples in Fig 3 (a), (b) and (d). That is not to stay that these examples are not interesting. On the contrary, I think they are very informative and interesting. But nonetheless, the authors should avoid such unsubstantiated remarks. 2- Unclear assumptions in Sec 2.2. a. Line 116, the Paragraph entitled "Update direction in Adam is close to "sign descent". I could not follow the reasoning in this paragraph. Could the authors please clearly state what assumptions they are relying in this paragraph. For instance in eq. 5 it should be clearly stated that these calculations hold when g_t is drawn from a stationary distribution. In this case we have, after including the bias correction rescaling for v_t, that E[v_t] = E[g_t^2] = Var(g_t) + (E[g_t])^2 Clarifying this is important since on line 129 it is stated "For Fig. 3(s), to match the assumption in Sec. 2.2., we set \beta_1 = \beta_2 = 0.3." I could not understand these parameter choices since it was clear what are the underlying assumptions of Sec 2.2. b. Paragraph on line 121 "when the variance of g_t is the same for all parameters..." By parameters, are you referring to the coordinates of g_t? Finally, may I confirm the conclusion of this paragraph is "under assumptions ? the Adam method is equivalent/approximately signed gradient descent which is not good because this is different than gradient descent"? I recommend that this paragraph be substantially re-written or removed. c. "Suppose the algorithm runs for a long time, so the EMA is very close to the expected value:" Confusing since you are running gradient descent then analysing m_t and v_t as if they were calculated on the iterates of gradient descent. Is this the case? 3- Theorem 2.1. > Problem is not defined. It seems here the problem should be defined to min_{\theta \in \mathcal{F}} f(\theta). This needs to be stated. Accordingly a projected step is added to Adabelief to tackle this constrained problem. This also needs to be stated in the theorem. > \beta_{1t} is not defined, nor is it roles explained. From checking the details of the proof, it appears that \beta_{1t} was meant to be the the momentum coefficient of m_t on the t-th iteration. This has to be clarified. > The significance of the result in Corollary should be clarified by bounding s_{t,i} and showing that \sum_{I=1}^d s_{t,i}^{1/2} < constant.

Correctness: The method and the empirical methodology appear correct. Due to the missing notation, problem definition and clarifications for Theorem 2.1, it was not possible to verify the correctness of the proof of Theorem 2.1

Clarity: The paper is generally well written, with a just a few paragraphs that are hard to parse due to missing assumptions, and some unsubstantiated statements.

Relation to Prior Work: The paper includes references to many recent developments and new variants of Adam and contrasts the new method to these alternatives.

Reproducibility: Yes

Additional Feedback: ******* UPDATE AFTER RESPONSE ********* Thank you for addressing my seven questions. I highly encourage the authors to incorporate the following improvements. I am also increasing my score to 7 under the understanding that the authors will properly address these points in their revision. Overall, I think it's good work and a good idea! 1. Add the clarifications on the assumptions in Section 2.2 2. Revise the connection with 2nd order information/curvature. I agree with R3 that the connection between Adabelief and 2nd curvature is not clear. In fact, I believe it to be misleading, and the benefits from Adabelief instead are due to using a centered 2nd momentum estimator. In particular, Figure 1 only really supports the use of |g_t-g_{t-1}| on the numerator. Which is not what is used in Adabelief. Indeed, even in the full batch this is not the numerator of Adabelief. Rather, using |g_t-g_{t-1}| in the denominator gives evidence that one should be using the secant equation to set the stepsize of each parameter (or coordinate). But this is not what Adabelief does. 3. Please introduce the notation used in Theorems and corollaries in Sec 2.2 ************************ Here I list some minor comments for the authors. I encourage the authors to consider these comments, but they do not need to address these comments in their rebuttal a. Videos slow convergence of Adam. I have one concern about these illustrations. Since the same parameter setting are used for Adam and Adabelief, Adam will effectively take a smaller step since (g_t -m_t)^2 < g_t^2. Thus I'm concerned that the apparent benefits of Adabelief in these videos is due to simply using an effective larger stepwise. Indeed, in general Adabelief can take larger step sizes than Adam. Is this why it performs better? b. Training stability: In several places in the paper, in particular the paragraph starting on line 223, the authors refer to the "stability of optimizers". What is meant here by stability? In the case of training GANs, some methods such as SGD can cycle. Is this cycling related to what the authors refer to a stability? c. Naming |\Delta \theta_t^i| as the ith step size is confusing for optimizers, since step size is a synonym for learning rate. Instead |Delta \theta_t^i| might be called the ith coordinate of the update direction. Furthermore, in line 98 you refer to |\Delta \theta_t| as the update direction. d. Is such a statement really necessary? : dine 181 " It is possible that the above bounds are loose; we will try to derive a tighter bound in the future." e. Unsubstantiated connection to the Hessian: To sum up, AdaBelief scales the update direction by the change in gradient, which is related to the Hessian. Therefore, AdaBelief considers curvature information and performs better than Adam. f. Figure 2, which optimizer does line 106 and this description "The optimizer oscillates in the y direction" refer to? It appears to be gradient descent with a fixed step size. g. Supplementary material. Why is there an additional epsilon in the bias correction step applied to s_t in Algorithm 1?


Review 2

Summary and Contributions: This paper shows a simple but interesting modification for Adam, uses 2D cases to intuitively explain the modification. Theoretical results are given and experiments are very promising.

Strengths: This paper claims to obtain speed, stability and generalization ability simutaneously. The modifican of ADAM is rather straightforward to implement, but the empirical results are very interesting: the test error can consistently improve almost 1 percent by just one line modification.

Weaknesses: I believe the intuitive explanation in Section 2.2 is not some kind of explanation, but a description for the phenomenon. The 1D, 2D results may not be generalized to the high dimensional setting. If the author hopes to claim the exploitation of curve information, the sound way is to show how the adabelief algorithm approximate the Hessian. The theoretical analysis is necessary but somewhat expected. In fact, I can not find any adavantage of AdaBelief in these theoretical results. If the s_{t,i} can be upper bounded better than the counterpart of Adam, then I believe the analysis has valuable justification.

Correctness: I believe the correctness of the results in this paper.

Clarity: Well written.

Relation to Prior Work: Clearly discussed.

Reproducibility: Yes

Additional Feedback: As I understand, in the deterministic setting, the resulted algorithm uses the EMA of gradient difference to approximate Hessian diag. If it is right, please clarify it. Meanwhile, in the stochastic setting, how we claim the approximation is reasonable? In my opinion, the regret analysis is not that useful as it is just routine that we need to do but no useful information can be obtained from it. I also a little doubt about using the nonsmooth 2D case to explain something related to Hessian. It is better to show some substaintial improvement of this approach at least in the deterministic setting.


Review 3

Summary and Contributions: The authors propose a centred variant of Adam, where the moving-average mean gradient is subtracted from the g^2 normaliser.

Strengths: Simple method that is in their hands is surprisingly effective.

Weaknesses: While the evaluations seem thorough, my concerns are: 1) The method is oversold: the change from raw second-moment to variance of the gradient is obvious-to-try, and potentially deserves a more descriptive name like "Centered Adam" rather than the slightly silly "AdaBelief". It is still interesting that it works, and the plots are convincing. 2) The intution is a bit belaboured: Adam slows down learning in directions with large *average* gradients, which isn't necessarily a good idea. This subtracts the bias, and therefore does not. 3) The notion of "belief" doesn't make much sense. They say "Intuitively, 1/sqrt(s_t) is the “belief” in the observation: viewing m_t as the prediction of the gradient, if g_t deviates much from m_t, we have weak belief in g_t, and take a small step; if g_t is close to the prediction m_t, we have a strong belief in g_t, and take a large step." This is problematic because first, it implies that the belief is something we compute on a per-gradient basis, whereas it is really something that is computed as a sum of past gradients, and second, it is true of the original Adam anyway --- at least as we approach the optimum, where we expect the minibatch noise to dominate the mean gradient. 4)As regards prior art, it is worth looking at two papers: Graves 2013. Generating Sequences With Recurrent Neural Networks. They do a centered version of RMSprop, which is very similar to the proposed method. Aitchison 2018. Bayesian filtering unifies adaptive and non-adaptive neural network optimization methods. They formulate learning in a NN as a Bayesian inference problem and develop a method "AdaBayes". This is worth noting while beliefs are what we reason about when doing Bayes theorem, "AdaBelief" has nothing to do with "AdaBayes". 5) Figure 3 seems a bit strange. AdaBelief seems to track Adam closely, and anyway I'd expect there to be strong hyperparameter dependence. I am willing to increase my score if the response can convince me that these issues have been addressed, which is possible as many of them concern the writing/presentation, as opposed to the results.

Correctness: Yes.

Clarity: Yes.

Relation to Prior Work: Modulo the prior art discussed above, yes.

Reproducibility: Yes

Additional Feedback: The response has not changed my view on these issues, though I take the point that the experimental evaluations are thorough.

[Author Response · NeurIPS 2020]

We thank all reviewers for comments. We are glad to see our work commented as "promising"(R3), "effective"(R6),
supported with "strong experimental results" and "intuitive justification"(R1). We address their concerns below.

**Response to R1**
**Q1. Writing**    We'll rephrase remarks, e.g."Examples give hints to local behavior of optimizers in deep learning".
**Q2.a Assumptions**    We list assumptions (1)-(3) as below:
–(1) assume $g_t$ is drawn from a stationary distribution, hence after bias correction, $\mathbb{E}v_t = (\mathbb{E}g_t)^2 + \mathbf{Var}g_t$.
–(2) low-noise assumption, $(\mathbb{E}g_t)^2 \gg \mathbf{Var}g_t$, hence we have $\mathbb{E}g_t/\sqrt{\mathbb{E}v_t} \approx \mathbb{E}g_t/\sqrt{(\mathbb{E}g_t)^2} = sign(\mathbb{E}g_t)$.
–(3) low-bias assumption, $\beta_1^t$ ($\beta_1$ to the power of $t$) is small. $m_t$ as an estimator of $\mathbb{E}g_t$ has bias $\beta_1^t\mathbb{E}g_t$, as in [1].
Numerically, we need a small $\beta$ (e.g 0.3) or large $t$. We also tried default $\beta$ with large $t$, results similar to Fig.3(d).
**Q2.b Conclusion**    ("Parameter" refer to coordinates of $g_t$) Under above assumptions, Adam is close to sign-descent,
which hurts performance , similar results explained in [3] (e.g. Lemma 2&3). We will rephrase as suggested.
**Q2.c Analysis setting, line 107**    By "run a long time", we refer to a large $t$, hence $\beta_1^t$ is small, and assumption (a.3) is
satisfied. $m_t, v_t$ are calculated strictly following Adam and updated with iterates, NOT post-hoc analysis of SGD.
**Q3.a Notations**    Our notations strictly follow the convention in [1,2]. We will add missing notations to Sec2.1 and
2.3. We use $(\beta_{1t}, \beta_{2t})$ to denote the momentum for $m_t$ and $v_t$ respectively at step $t$, and typically set as constant (e.g.
$\beta_{1t} = \beta_1, \beta_{2t} = \beta_2, \forall t \in \{1, 2, ...T\}$, where $T$ is the total number of steps). Note that $\beta_{1t} \neq \beta_1^t$, $\beta_1^t$ is $\beta_1$ to the power
$t$. As in Algo. 1 in Appendix A, we use $\widehat{s_t}$ and $\widehat{m_t}$ to denote the bias-corrected version of $s_t$ and $m_t$ respectively.
**Q3.b Optimization problem**    Strictly following the convention in [1,2], for deterministic problems, the problem to
be optimized is $\min_{\theta \in \mathcal{F}} f(\theta)$; for online optimization, the problem is $\min_{\theta \in \mathcal{F}} \sum_{t=1}^{T} f_t(\theta)$, where $f_t$ can be interpreted
as "loss of the model with the chosen parameters in $t$-th step" [2].
**Q3.c Projection step**    A detailed version of our method with projection step is in Appendix A. Our proof already
considers projection, see Lemma 0.1 and Formula.(1) in Appendix B.
**Q3.d Corollary 2.1.1**    **(1)** Similar to Theorem 4.1 in [1] and corollary 1 in [2], where the term $\sum_{i=1}^{d} v_{T,i}^{1/2}$ exists, we
have $\sum_{i=1}^{d} s_{T,i}^{1/2}$. Without further assumption, $\sum_{i=1}^{d} s_{T,i}^{1/2} < dG_\infty$ since $||g_t - m_t||_\infty < G_\infty$ as assumed in Theorem
2.1, and $dG_\infty$ is constant. **(2)** The literature [1,2,5] exerts a stronger assumption that $\sum_{i=1}^{d} T^{1/2}v_{T,i}^{1/2} \ll dG_\infty T^{1/2}$.
Our assumption could be similar or weaker, because $\mathbb{E}s_t = \mathbf{Var}g_t \leq \mathbb{E}g_t^2 = \mathbb{E}v_t$, then get better regret than $O(T^{1/2})$.
**Response to additional comments**
**(a)** No, see response to Q2 of R6. **(b)** Yes. It's related to "cycle" in theory, and "mode collapse" in practice. **(e)**
see response to R3 below. **(f)** We refer to all three optimizers. Fig2 is illustrative; rigorously, oscillation amplitude
in y-axis decreases, but gradient is independent of the distance to axis for L1 loss, hence our analysis holds for
both fixed-step-size and decreasing-step-size. **(g)** We absorb $\epsilon$ into $s_t$ in theoretical analysis, in implementation
we add $\epsilon$ to match assumption $s_t > c > 0$ in Theorem 2.1 ($c \geq \epsilon > 0$). AdaBelief is robust to $\epsilon$, as Fig.4 in Appendix.

**Response to R3**    We only claim AdaBelief is related to Hessian but not necessarily a good approximation, mainly
because: (1) in Newton method, the update is $H^{-1}\nabla f$, using $diag(H)^{-1}$ to approximate $H^{-1}$ may cause problems.
It might be better to directly approximate $H^{-1}\nabla f$ rather than approximating $H$ as $diag(H)$. (2) omitting the
effect of EMA, $g_t - g_{t-1} \approx H\Delta\theta_t$, where $\Delta\theta_t$ is the update of parameter; in other words, $g_t - g_{t-1}$ approximates
the product of $H$ with a direction $\Delta\theta_t$, rather than approximating $diag(H)$. (3) Adam-type methods use $1/\sqrt{v_t}$, which is
approximation to $H^{-1/2}$ rather than $H^{-1}$. We'll work on a tighter bound from Hessian perspective in future work.

**Response to R6**
**Q1. Simplicity**    Our method is "simple but effective" (R6). To our knowledge, it's novel and uninvestigated before.
**Q2. Comparison with Adam**    We address R6's concern that the success of AdaBelief stems largely from an
effectively larger stepsize. We argue that this is not the case. **(1)** As in Fig.5 in Appendix, for various learning rates,
AdaBelief consistently outperforms the best choice of Adam, including when Adam uses a much larger lr than
AdaBelief. Validating the performance improvement of Adabelief does not solely come from larger stepsize. **(2)** when
$sign(g_t) \neq sign(m_t)$ (e.g. due to noise in $g_t$) hence $(g_t - m_t)^2 > g_t^2$, AdaBelief can take a smaller step than Adam.
**Q3. Name of our method**    **(1)** We use the word "belief" in a colloquial sense to refer to the amount by which the
observed gradient $g_t$ deviates from its exponential moving average $m_t$(viewed as approximated expected gradient).
Updates in AdaBelief are "per-gradient" and element-wise, similar to Adam[1] and AdaBayes[6] where they all depend
on history gradients implicitly due to the momentum and iterative update. **(2)** R6 argues intuition for AdaBelief holds
for Adam, which is not true. AdaBelief resembles Adam when $(\mathbb{E}g_t)^2 \ll \mathbf{Var}g_t$. When $(\mathbb{E}g_t)^2 \gg \mathbf{Var}g_t$ Adam is close
to "sign-descent" and affects accuracy, explained in Sec.2.2 of our paper and [3]; while AdaBelief overcomes this.
**Q4. Prior work**    **(1)** The denominator in [4] is $(v_t - m_t^2)^{1/2}$, could result in numerical errors (e.g. $v_t - m_t^2 < 0$),
as the authors mentioned. AdaBelief uses $[EMA((g_t - m_t)^2)]^{1/2}$ as denominator, guaranteed to be valid operation,
and trains LSTM successfully without numerical issues. Compared with [4], we provide extensive theoretical and
experimental validations. **(2)** [6] is completely different, "AdaBelief has nothing to do with AdaBayes" (by R6).
**Q5.** In Fig.3, AdaBelief uses same hyperparameters as Adam thus have similar trajectories, but reaches optima faster.

**References** [1] Kingma et. al, Adam: A method for stochastic optimization [2] Reddi et. al, On the convergence of Adam and
beyond. [3] Lukas et. al, Dissecting adam: The sign, magnitude and variance of stochastic gradients [4] Graves et. al, Generating
sequences with recurrent neural networks [5] Duchi, Adaptive subgradient methods for online learning and stochastic optimization
[6] Aitchison, Bayesian filtering unifies adaptive and non-adaptive neural network optimization methods


[Meta-Review · NeurIPS 2020]

All reviewers agree that the proposed method makes simple and effective change to the popular Adam algorithm, supported by strong empirical results and relatively standard convergence guarantees. Due to its simplicity, effectiveness and clear and convincing writing, the method has the potential of becoming a new standard method in deep learning. Therefore I recommend acceptance. The reviewers have concerns of somewhat unsubstantiated claims and oversold statements, but I believe these are relatively minor compared with the contribution. I urge the authors to carefully address these concerns in the revision.